# iBOT 🤖: Image BERT Pre-Training with Online Tokenizer

**Jinghao Zhou**[1]  **Chen Wei**[2]  **Huiyu Wang**[2]  **Wei Shen**[3]  **Cihang Xie**[4]  **Alan Yuille**[2]  **Tao Kong**[1]

[1]ByteDance  [2]Johns Hopkins University  [3]Shanghai Jiao Tong University  [4]UC Santa Cruz

## ABSTRACT

The success of language Transformers is primarily attributed to the pretext task of masked language modeling (MLM) (Devlin et al., 2019), where texts are first tokenized into semantically meaningful pieces. In this work, we study masked image modeling (MIM) and indicate the advantages and challenges of using a semantically meaningful visual tokenizer. We present a self-supervised framework iBOT that can perform masked prediction with an *online tokenizer*. Specifically, we perform self-distillation on masked patch tokens and take the teacher network as the online tokenizer, along with self-distillation on the class token to acquire visual semantics. The online tokenizer is jointly learnable with the MIM objective and dispenses with a multi-stage training pipeline where the tokenizer needs to be pretrained beforehand. We show the prominence of iBOT by achieving an **82.3%** linear probing accuracy and an **87.8%** fine-tuning accuracy evaluated on ImageNet-1K. Beyond the state-of-the-art image classification results, we underline emerging local semantic patterns, which helps the models to obtain strong robustness against common corruptions and achieve leading results on dense downstream tasks, *e.g.*, object detection, instance segmentation, and semantic segmentation. The code and models are publicly available at https://github.com/bytedance/ibot.

## 1 INTRODUCTION

Masked Language Modeling (MLM), which first randomly masks and then reconstructs a set of input tokens, is a popular pre-training paradigm for language models. The MLM pre-trained Transformers (Devlin et al., 2019) have demonstrated their scalability to large-capacity models and datasets, becoming a de-facto standard for lingual tasks. However, its potential for Vision Transformer (ViT), which recently started to revolutionize computer vision research (Touvron et al., 2021; Dosovitskiy et al., 2021), has been largely under-explored. Most popular unsupervised pre-training schemes in vision deal with the global views (Chen et al., 2021; Caron et al., 2021), neglecting images' internal structures, as opposed to MLM modeling local tokens. In this work, we seek to continue the success of MLM and explore Masked Image Modeling (MIM) for training better Vision Transformers such that it can serve as a standard component, as it does for NLP.

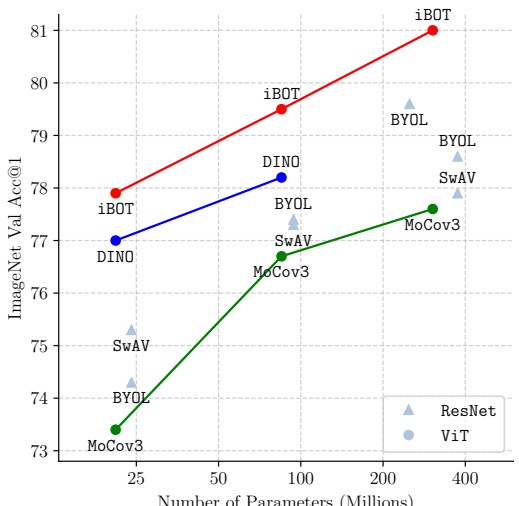

Figure 1: **Linear probing accuracy on ImageNet.** We compare iBOT with other unsupervised baselines.

One of the most crucial components in MLM is the *lingual tokenizer* which splits language into semantically meaningful tokens, *e.g.*, WordPiece (Wu et al., 2016) in BERT. Similarly, the crux of MIM lies in a proper design of *visual tokenizer*, which transforms the masked patches to supervisory

Email: Jinghao Zhou<jensen.zhoujh@gmail.com>, Tao Kong<kongtao@bytedance.com>.

signals for the target model, as shown in Fig. 2. However, unlike lingual semantics arising naturally from the statistical analysis of word frequency (Sennrich et al., 2016), visual semantics cannot be extracted such easily due to the continuous property of images. Empirically, visual semantics emerges progressively by bootstrapping online representation that enforces a similarity of distorted image views (He et al., 2020; Grill et al., 2020; Caron et al., 2020). This property intuitively indicates a multi-stage training pipeline, where we need to first train an off-the-shelf semantic-rich tokenizer before training the target model. However, since acquiring visual semantics is a common end for both the tokenizer and target model, a single-stage training pipeline where the tokenizer and target model can be jointly optimized awaits further exploration.

Previous works partially tackle the above challenges. Several works use identity mapping as the visual tokenizer, *i.e.*, predicting the raw pixel values (Pathak et al., 2016; Atito et al., 2021). Such paradigm struggles in semantic abstraction and wastes the capacity at modeling high-frequency details, yielding less competitive performance in semantic understanding (Liu et al., 2021a). Recently, BEiT (Bao et al., 2021) proposes to use a pre-trained discrete VAE (Ramesh et al., 2021) as the tokenizer. Though providing some level of abstraction, the discrete VAE is still found only to capture low-level semantics within local details (as observed by Tab. 9). Moreover, the tokenizer needs to be offline pre-trained with fixed model architectures and extra dataset (Ramesh et al., 2021), which potentially limits its adapativity to perform MIM using data from different domains.

To this end, we present iBOT 🦾, short for **i**mage **B**ERT pre-training with **O**nline **T**okenizer, a new framework that performs MIM with a tokenizer handling above-mentioned challenges favorably. We motivate iBOT by formulating the MIM as knowledge distillation (KD), which learns to distill knowledge from the tokenizer, and further propose to perform self-distillation for MIM with the help of twin teacher as online tokenizer. The target network is fed with a masked image while the online tokenizer with the original image. The goal is to let the target network recover each masked patch token to its corresponding tokenizer output. Our online tokenizer naturally resolves two major challenges. On the one hand, our tokenizer captures high-level visual semantics progressively learned by enforcing the similarity of cross-view images on class tokens. On the other hand, our tokenizer needs no extra stages of training as pre-processing setup since it is jointly optimized with MIM via momentum update.

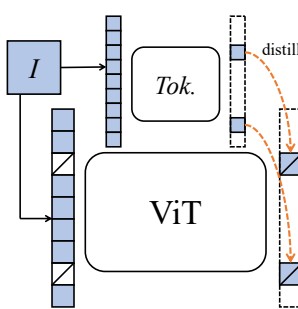

Figure 2: **Masked image modeling.** *I* denotes an image and *Tok.* denotes a visual tokenizer.

The online tokenizer enables iBOT to achieve excellent performance for feature representation. Specifically, iBOT advances ImageNet-1K classification benchmark under $k$-NN, linear probing and fine-tuning protocols to 77.1%, 79.5%, 84.0% with ViT-Base/16 respectively, which is 1.0%, 1.3%, 0.4% higher than previous best results. When pre-trained with ImageNet-22K, iBOT with ViT-L/16 achieves a linear probing accuracy of **82.3%** and a fine-tuning accuracy of **87.8%**, which is 1.0% and 1.8% higher than previous best results. Beyond that, the advancement is also valid when transferring to other datasets or under semi-supervised and unsupervised classification settings. Of particular interest, we have identified an emerging part-level semantics that can help the model with image recognition both on global and local scales. We identify that the semantic patterns learned in patch tokens, which sufficiently lack in the off-line tokenizer as in BEiT (Bao et al., 2021), helps the model to be advanced in linear classification and robustness against common image corruptions. When it is transferred to downstream tasks, we show that in downstream tasks related to image classification, object detection, instance segmentation, and semantic segmentation, iBOT surpasses previous methods with nontrivial margins. All of the evidence demonstrates that iBOT has largely closed the gap of masked modeling pre-training between language and vision Transformers.

## 2 PRELIMINARIES

### 2.1 MASKED IMAGE MODELING AS KNOWLEDGE DISTILLATION

Masked image modeling (MIM), which takes a similar formulation as MLM in BERT, has been proposed in several recent works (Bao et al., 2021; Tan et al., 2021). Specifically, for an image token sequence $x = \{x_i\}_{i=1}^N$, MIM first samples a random mask $m \in \{0, 1\}^N$ according to a prediction ratio $r$, where $N$ is the number of tokens. The patch token $x_i$ where $m_i$ being 1, denoted

as $\tilde{x} \triangleq \{x_i \mid m_i = 1\}$, are then replaced with a mask token $e_{\texttt{[MASK]}}$, yielding a corrupted image $\hat{x} \triangleq \{\hat{x}_i \mid (1 - m_i)x_i + m_i e_{\texttt{[MASK]}}\}_{i=1}^{N}$. MIM is to recover the masked tokens $\tilde{x}$ from the corrupted image $\hat{x}$, $i.e.$, to maximize: $\log q_{\boldsymbol{\theta}}(\tilde{x}|\hat{x}) \approx \sum_{i=1}^{N} m_i \cdot \log q_{\boldsymbol{\theta}}(x_i|\hat{x})$, where $\approx$ holds with an independence assumption that each masked token can be reconstructed separately. In BEiT (Bao et al., 2021), $q_{\boldsymbol{\theta}}$ is modelled as a categorical distribution and the task is to minimize

$$-\sum_{i=1}^{N} m_i \cdot P_{\boldsymbol{\phi}}(x_i)^{\mathrm{T}} \log P_{\boldsymbol{\theta}}(\hat{x}_i), \tag{1}$$

where $P(\cdot)$ transforms the input to a probability distribution over $K$ dimensions, and $\phi$ is parameters of a discrete VAE (Ramesh et al., 2021) that clusters image patches into $K$ categories and assigns each patch token a one-hot encoding identifying its category. We note this loss is formulated similarly to knowledge distillation (Hinton et al., 2015), where knowledge is distilled from a pre-fixed tokenizer parameterized by $\phi$ to current model parameterized by $\boldsymbol{\theta}$.

## 2.2 SELF-DISTILLATION

Self-distillation, proposed recently in DINO (Caron et al., 2021), distills knowledge not from posterior distributions $P_{\boldsymbol{\phi}}(x)$ but past iterations of model itself $P_{\boldsymbol{\theta}'}(x)$ and is cast as a *discriminative* self-supervised objective. Given the training set $\mathcal{I}$, an image $x \sim \mathcal{I}$ is sampled uniformly, over which two random augmentations are applied, yielding two distorted views $u$ and $v$. The two distorted views are then put through a teacher-student framework to get the predictive categorical distributions from the $\texttt{[CLS]}$ token: $v_t^{\texttt{[CLS]}} = P_{\boldsymbol{\theta}'}^{\texttt{[CLS]}}(v)$ and $u_s^{\texttt{[CLS]}} = P_{\boldsymbol{\theta}}^{\texttt{[CLS]}}(u)$. The knowledge is distilled from teacher to student by minimizing their cross-entropy, formulated as

$$\mathcal{L}_{\texttt{[CLS]}} = -P_{\boldsymbol{\theta}'}^{\texttt{[CLS]}}(v)^{\mathrm{T}} \log P_{\boldsymbol{\theta}}^{\texttt{[CLS]}}(u). \tag{2}$$

The teacher and the student share the same architecture consisting of a backbone $f$ ($e.g.$, ViT) and a projection head $h^{\texttt{[CLS]}}$. The parameters of the student network $\boldsymbol{\theta}$ are Exponentially Moving Averaged (EMA) to the parameters of teacher network $\boldsymbol{\theta}'$. The loss is symmetrized by averaging with another cross-entropy term between $v_s^{\texttt{[CLS]}}$ and $u_t^{\texttt{[CLS]}}$.

## 3 IBOT

We motivate our method by identifying the similar formulation of Eq. (1) and Eq. (2). A visual tokenizer parameterized by online $\boldsymbol{\theta}'$ instead of pre-fixed $\phi$ thus arises naturally. In this section, we present iBOT, casting self-distillation as a *token-generation* self-supervised objective and perform MIM via self-distillation. We illustrate the framework of iBOT in Fig. 3 and demonstrate the pseudo-code in Appendix A. In Sec. 3.2, we briefly introduce the architecture and pre-training setup.

## 3.1 FRAMEWORK

First, we perform blockwise masking (Bao et al., 2021) on the two augmented views $u$ and $v$ and obtain their masked views $\hat{u}$ and $\hat{v}$. Taking $\hat{u}$ as an example for simplicity, the student network outputs for the masked view $\hat{u}$ projections of its patch tokens $\hat{u}_s^{\mathrm{patch}} = P_{\boldsymbol{\theta}}^{\mathrm{patch}}(\hat{u})$ and the teacher network outputs for the non-masked view $u$ projections of its patch tokens $u_t^{\mathrm{patch}} = P_{\boldsymbol{\theta}'}^{\mathrm{patch}}(u)$. We here define the training objective of MIM in iBOT as

$$\mathcal{L}_{\mathrm{MIM}} = -\sum_{i=1}^{N} m_i \cdot P_{\boldsymbol{\theta}'}^{\mathrm{patch}}(u_i)^{\mathrm{T}} \log P_{\boldsymbol{\theta}}^{\mathrm{patch}}(\hat{u}_i). \tag{3}$$

We symmetrize the loss by averaging with another CE term between $\hat{v}_s^{\mathrm{patch}}$ and $v_t^{\mathrm{patch}}$.

The backbone together with the projection head of teacher network $h_t^{\mathrm{patch}} \circ f_t$ is, therefore, a visual tokenizer that generates online token distributions for each masked patch token. The tokenizer used in iBOT is jointly learnable to MIM objective without a need of being pre-trained in an extra stage, a bonus feature of which is now its domain knowledge can be distilled from the current dataset rather than fixed to the specified dataset.

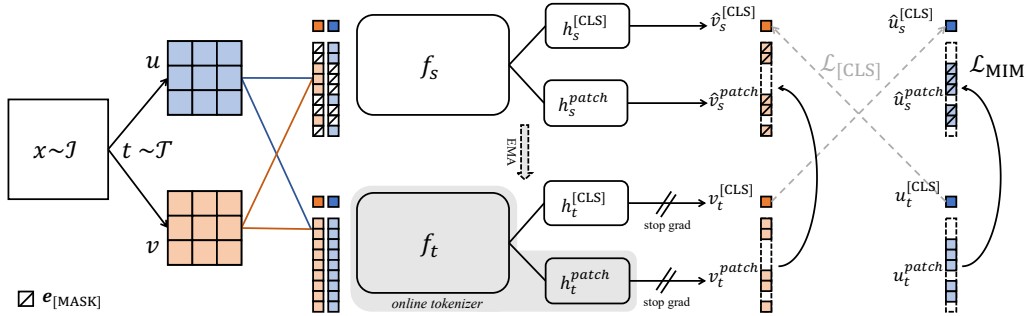

Figure 3: **Overview of iBOT framework, performing masked image modeling with an *online tokenizer*.** Given two views $u$ and $v$ of an image $x$, each view is passed through a teacher network $h_t \circ f_t$ and a student network $h_s \circ f_s$. iBOT minimizes two losses. The first loss $\mathcal{L}_{\texttt{[CLS]}}$ is self-distillation between cross-view $\texttt{[CLS]}$ tokens. The second loss $\mathcal{L}_{\mathrm{MIM}}$ is self-distillation between in-view patch tokens, with some tokens masked and replaced by $e_{\texttt{[MASK]}}$ for the student network. The objective is to reconstruct the masked tokens with the teacher networks' outputs as supervision.

To ensure that the online tokenizer is semantically-meaningful, we perform self-distillation on $\texttt{[CLS]}$ token of cross-view images such that visual semantics can be obtained via bootstrapping, as achieved by the majority of the self-supervised methods (He et al., 2020; Grill et al., 2020; Caron et al., 2021). In practice, iBOT works with $\mathcal{L}_{\texttt{[CLS]}}$ in Eq. (2) proposed in DINO (Caron et al., 2021), except that now we have $\hat{u}_s^{\texttt{[CLS]}}$ instead of $u_s^{\texttt{[CLS]}}$ as input for the student network. To further borrow the capability of semantics abstraction acquired from self-distillatin on $\texttt{[CLS]}$ token, we share the parameters of projection heads for $\texttt{[CLS]}$ token and patch tokens, *i.e.*, $h_s^{\texttt{[CLS]}} = h_s^{\mathrm{patch}}$, $h_t^{\texttt{[CLS]}} = h_t^{\mathrm{patch}}$. We empirically find that it produces better results than using separate heads.

Unlike tokenized words whose semantics are almost certain, image patch is ambiguous in its semantic meaning. Therefore, tokenization as one-hot discretization can be sub-optimal for images. In iBOT, we use the token distribution after softmax instead of the one-hot token id as a supervisory signal, which plays an important role in iBOT pre-training as shown in Tab. 18.

## 3.2 IMPLEMENTATION

**Architecture.** We use the Vision Transformers (Dosovitskiy et al., 2021) and Swin Transformers (Liu et al., 2021b) with different amounts of parameters, ViT-S/16, ViT-B/16, ViT-L/16, and Swin-T/$\{7,14\}$ as the backbone $f$. For ViTs, /16 denotes the patch size being 16. For Swins, /$\{7,14\}$ denotes the window size being 7 or 14. We pre-train and fine-tune the Transformers with 224-size images, so the total number of patch tokens is 196. The projection head $h$ is a 3-layer MLPs with $l_2$-normalized bottleneck following DINO (Caron et al., 2021). Towards a better design to acquire visual semantics, we studied different sharing strategies between projection heads $h^{\texttt{[CLS]}}$ and $h^{\mathrm{patch}}$, considering that semantics obtained in distillation on $\texttt{[CLS]}$ token helps the training of MIM on patch tokens. We empirically find that sharing the entire head prompts the best performance. We set the output dimension of the shared head to 8192.

**Pre-Training Setup.** We by default pre-train iBOT on ImageNet-1K (Deng et al., 2009) training set with AdamW (Loshchilov & Hutter, 2019) optimizer and a batch size of 1024. We pre-train iBOT with ViT-S/16 for 800 epochs, ViT-B/16 for 400 epochs, ViT-L/16 for 250 epochs, and Swin-T/$\{7,14\}$ for 300 epochs. We also pre-train on ImageNet-22K training set with ViT-B/16 for 80 epochs and ViT-L/16 for 50 epochs. The learning rate is linearly ramped up during the first 10 epochs to its base value scaled with the total batch size: $\mathrm{lr} = 5e^{-4} \times \mathrm{batch\_size}/256$. We use random MIM, with prediction ratio $r$ set as 0 with a probability of 0.5 and uniformly sampled from range [0.1, 0.5] with a probability of 0.5. We sum $\mathcal{L}_{\texttt{[CLS]}}$ and $\mathcal{L}_{\mathrm{MIM}}$ up without scaling.

## 4 EXPERIMENT

We first transfer iBOT to downstream tasks, following the standard evaluation protocols adopted in prior arts, the details of which are delayed in Appendix C. We then study several interesting

Table 1: $k$-**NN and linear probing on ImageNet-1K**. [†] denotes using selective kernel. [‡] denotes pre-training on ImageNet-22K.

| Method | Arch. | Par. | im/s | Epo.[1] | $k$-NN | Lin. |
|---|---|---|---|---|---|---|
| *SSL big ResNets* | | | | | | |
| MoCov3 | RN50 | 23 | 1237 | 1600 | - | 74.6 |
| SwAV | RN50 | 23 | 1237 | 2400 | 65.7 | 75.3 |
| DINO | RN50 | 23 | 1237 | 3200 | 67.5 | 75.3 |
| BYOL | RN200w2 | 250 | 123 | 2000 | 73.9 | 79.6 |
| SCLRv2 | RN152w3[†] | 794 | 46 | 2000 | 73.1 | 79.8 |
| *SSL Transformers* | | | | | | |
| MoCov3 | ViT-S/16 | 21 | 1007 | 1200 | - | 73.4 |
| MoCov3 | ViT-B/16 | 85 | 312 | 1200 | - | 76.7 |
| SwAV | ViT-S/16 | 21 | 1007 | 2400 | 66.3 | 73.5 |
| DINO | ViT-S/16 | 21 | 1007 | 3200 | 74.5 | 77.0 |
| DINO | ViT-B/16 | 85 | 312 | 1600 | 76.1 | 78.2 |
| EsViT | Swin-T/7 | 28 | 726 | 1200 | 75.7 | 78.1 |
| EsViT | Swin-T/14 | 28 | 593 | 1200 | 77.0 | 78.7 |
| iBOT | ViT-S/16 | 21 | 1007 | 3200 | 75.2 | 77.9 |
| iBOT | Swin-T/7 | 28 | 726 | 1200 | 75.3 | 78.6 |
| iBOT | Swin-T/14 | 28 | 593 | 1200 | 76.2 | 79.3 |
| iBOT | ViT-B/16 | 85 | 312 | 1600 | 77.1 | 79.5 |
| iBOT | ViT-L/16 | 307 | 102 | 1200 | **78.0** | 81.0 |
| iBOT [‡] | ViT-L/16 | 307 | 102 | 200 | 72.9 | **82.3** |

Table 2: **Fine-tuning on ImageNet-1K.**

| Method | Arch. | Epo.[1] | Acc. |
|---|---|---|---|
| Rand. | ViT-S/16 | - | 79.9 |
| MoCov3 | ViT-S/16 | 600 | 81.4 |
| DINO | ViT-S/16 | 3200 | 82.0 |
| iBOT | ViT-S/16 | 3200 | **82.3** |
| Rand. | ViT-B/16 | - | 81.8 |
| MoCov3 | ViT-B/16 | 600 | 83.2 |
| BEiT | ViT-B/16 | 800 | 83.4 |
| DINO | ViT-B/16 | 1600 | 83.6 |
| iBOT | ViT-B/16 | 1600 | **84.0** |
| MoCov3 | ViT-L/16 | 600 | 84.1 |
| iBOT | ViT-L/16 | 1000 | 84.8 |
| BEiT | ViT-L/16 | 800 | **85.2** |

Table 3: **Fine-tuning on ImageNet-1K.** Pre-training on ImageNet-22K.

| Method | Arch. | Epo.[1] | Acc. |
|---|---|---|---|
| BEiT | ViT-B/16 | 150 | 83.7 |
| iBOT | ViT-B/16 | 320 | **84.4** |
| BEiT | ViT-L/16 | 150 | 86.0 |
| iBOT | ViT-L/16 | 200 | 86.6 |
| iBOT | ViT$_{512}$-L/16 | 200 | **87.8** |

properties of Transformers pre-trained with iBOT. Finally, we give a brief ablation study on the crucial composing of iBOT.

## 4.1 CLASSIFICATION ON IMAGENET-1K

We consider five classification protocols on ImageNet-1K: $k$-NN, linear probing, fine-tuning, semi-supervised learning, and unsupervised learning.

$k$-**NN and Linear Probing.** To evaluate the quality of pre-trained features, we either use a $k$-nearest neighbor ($k$-NN) classifier or a linear classifier on the frozen representation. We follow the evaluation protocols in DINO (Caron et al., 2021). For $k$-NN evaluation, we sweep over different numbers of nearest neighbors. For linear evaluation, we sweep over different learning rates. In Tab. 1, our method reaches a linear probing accuracy 77.9% with ViT-S/16, a linear probing accuracy 79.5% with ViT-B/16, and a $k$-NN accuracy 78.0% and linear probing accuracy 81.0% with ViT-L/16, achieving state-of-the-art performance. With Swin-T/{7,14}, iBOT achieves a linear probing accuracy of 78.6% and 79.3% respectively. With ViT-L/16 and ImageNet-22K as pre-training data, iBOT further achieves a linear probing accuracy **82.3%**, surpassing previous state of the art, 81.3% with Swin-B/14 by EsViT (Li et al., 2021a). A linear probing accuracy of 79.5% with ViT-B/16 is comparable to 79.8% by SimCLRv2 with RN152 $(3\times)^{\dagger}$ but with $10\times$ less parameters. We underline that the performance gain over DINO gets larger (0.9% w/ ViT-S versus 1.3% w/ ViT-B) with more parameters, suggesting iBOT is more scalable to larger models.

**Fine-Tuning.** We study the fine-tuning on ImageNet-1K and focus on the comparison with self-supervised methods for Transformers and its supervised baseline (*Rand.*) (Touvron et al., 2021). As shown in Tab. 2, iBOT achieves an 82.3%, 84.0%, and 84.8% top-1 accuracy with ViT-S/16, ViT-B/16, and ViT-L/16, respectively. As shown in Tab. 3, iBOT pre-trained with ImageNet-22K achieves 84.4% and 86.6% top-1 accuracy with ViT-B/16 and ViT-L/16, respectively, outperforming ImageNet-22K pre-trained BEiT by 0.7% and 0.6%. When fine-tuned on an image size of 512, we achieve **87.8%** accuracy. We note that, with ViT-L/16, iBOT is 0.4% worse than BEiT using 1K data but 0.6% better using 22K data. This implies that iBOT requires more data to train larger model.

**Semi-Supervised and Unsupervised Learning.** For semi-supervised learning, we focus our comparison with methods following the *unsupervised pre-train, supervised fine-tune* paradigm. As

---

[1]Effective pre-training epochs accounting for actual trained images/views. See Appenix B for details.

Table 4: **Semi-supervised learning on ImageNet-1K.** 1% and 10% denotes label fraction. SD denotes self-distillation.

| Method | Arch. | 1% | 10% |
|---|---|---|---|
| SimCLRv2 | RN50 | 57.9 | 68.1 |
| BYOL | RN50 | 53.2 | 68.8 |
| SwAV | RN50 | 53.9 | 70.2 |
| SimCLRv2+SD | RN50 | 60.0 | 70.5 |
| DINO | ViT-S/16 | 60.3 | 74.3 |
| iBOT | ViT-S/16 | **61.9** | **75.1** |

Table 5: **Unsupervised learning on ImageNet-1K.** [†] denotes $k$-means clustering on frozen features.

| Method | Arch. | ACC | ARI | NMI | FMI |
|---|---|---|---|---|---|
| Self-label[†] | RN50 | 30.5 | 16.2 | 75.4 | - |
| InfoMin[†] | RN50 | 33.2 | 14.7 | 68.8 | - |
| SCAN | RN50 | 39.9 | 27.5 | 72.0 | - |
| DINO | ViT-S/16 | 41.4 | 29.8 | 76.8 | 32.8 |
| iBOT | ViT-S/16 | **43.4** | **32.8** | **78.6** | **35.6** |

Table 6: **Object detection (Det.) & instance segmentation (ISeg.) on COCO and Semantic segmentation (Seg.) on ADE20K.** We report the results of ViT-S/16 (left) and ViT-B/16 (right). Seg.[†] denotes using a linear head for semantic segmentation.

| Method | Arch. | Param. | Det. $AP^b$ | ISeg. $AP^m$ | Seg. mIoU | | Method | Det. $AP^b$ | ISeg. $AP^m$ | Seg.[†] mIoU | Seg. mIoU |
|---|---|---|---|---|---|---|---|---|---|---|---|
| Sup. | Swin-T | 29 | 48.1 | 41.7 | 44.5 | | Sup. | 49.8 | 43.2 | 35.4 | 46.6 |
| MoBY | Swin-T | 29 | 48.1 | 41.5 | 44.1 | | BEiT | 50.1 | 43.5 | 27.4 | 45.8 |
| Sup. | ViT-S/16 | 21 | 46.2 | 40.1 | 44.5 | | DINO | 50.1 | 43.4 | 34.5 | 46.8 |
| iBOT | ViT-S/16 | 21 | **49.4** | **42.6** | **45.4** | | iBOT | **51.2** | **44.2** | **38.3** | **50.0** |

shown in Tab. 4, iBOT advances DINO by 1.6% and 0.8% using 1% and 10% data, respectively, suggesting a higher label efficiency. For unsupervised learning, we use standard evaluation metrics, including accuracy (ACC), adjusted random index (ARI), normalized mutual information (NMI), and Fowlkes-Mallows index (FMI). We compare our methods to SimCLRv2 (Chen et al., 2020b), Self-label (Asano et al., 2020), InfoMin (Tian et al., 2020), and SCAN (Van Gansbeke et al., 2020). As shown in Tab. 5, we achieve a **32.8%** NMI, outperforming the previous state of the art by 1.8%, suggesting MIM helps the model learn stronger visual semantics on a global scale.

## 4.2 DOWNSTREAM TASKS

**Object Detection and Instance Segmentation on COCO.** Object detection and instance segmentation require simultaneous object location and classification.We consider Cascade Mask R-CNN (Cai & Vasconcelos, 2019; He et al., 2017) that produces bounding boxes and instance masks simultaneously on COCO dataset (Lin et al., 2014). Several recent works (Liu et al., 2021b; Wang et al., 2021a) proposes Vision Transformers that suit dense downstream tasks. To compare, we include the results of supervised Swin-T (Liu et al., 2021b) which shares approximate parameter numbers with ViT-S/16 and its self-supervised counterpart MoBY (Xie et al., 2021a) in Tab. 6. iBOT improves ViT-S's $AP^b$ from 46.2 to 49.4 and $AP^m$ from 40.1 to 42.6, surpassing both supervised Swin-T and its self-supervised counterpart by a nontrivial margin. With ViT-B/16, iBOT achieves an $AP^b$ of 51.2 and an $AP^m$ of 44.2, surpassing previous best results by a large margin.

**Semantic Segmentation on ADE20K.** Semantic segmentation can be seen as a pixel-level classification problem. We mainly consider two segmentation settings on ADE20K dataset (Zhou et al., 2017). First, similar to linear evaluation protocol in classification, we evaluate on the fixed patch features and only fine-tune a linear layer, which gives us a more explicit comparison of the quality of representations. Second, we use the task layer in UPerNet (Xiao et al., 2018) and fine-tune the entire network. From Tab. 6, we can see that iBOT advances its supervised baseline with ViT-S/16 with a large margin of 0.9 on mIoU, surpassing Swin-T. With ViT-B/16, iBOT advances previous best methods DINO by 3.2 on mIoU with UperNet. We notice a performance drop of BEiT using linear head, indicating BEiT's features lack local semantics. As analyzed later, the property of strong local semantics induces a 2.9 mIoU gain compared to the supervised baseline with a linear head.

**Transfer Learning.** We study transfer learning where we pre-train on ImageNet-1K and fine-tune on several smaller datasets.We follow the training recipe and protocol used in (Dosovitskiy et al., 2021). The results are demonstrated in Tab. 7. While the results on several datasets (*e.g.*, CIFAR10, CIFAR100, Flowers, and Cars) have almost plateaued, iBOT consistently performs favorably against other SSL frameworks, achieving state-of-the-art transfer results. We observe greater performance

Table 7: **Transfer learning by fine-tuning pre-trained models on different datasets.** We report Top-1 accuracy of ViT-S/16 (left) and ViT-B/16 (right).

| Method | $Cif_{10}$ | $Cif_{100}$ | $iNa_{18}$ | $iNa_{19}$ | Flwrs | Cars | Method | $Cif_{10}$ | $Cif_{100}$ | $iNa_{18}$ | $iNa_{19}$ | Flwrs | Cars |
|--------|-----------|-------------|-----------|-----------|-------|------|--------|-----------|-------------|-----------|-----------|-------|------|
| Rand. | 99.0 | 89.5 | 70.7 | 76.6 | 98.2 | 92.1 | Rand. | 99.0 | 90.8 | 73.2 | 77.7 | 98.4 | 92.1 |
| BEiT | 98.6 | 87.4 | 68.5 | 76.5 | 96.4 | 92.1 | BEiT | 99.0 | 90.1 | 72.3 | 79.2 | 98.0 | 94.2 |
| DINO | 99.0 | 90.5 | 72.0 | 78.2 | 98.5 | 93.0 | DINO | 99.1 | 91.7 | 72.6 | 78.6 | 98.8 | 93.0 |
| iBOT | **99.1** | **90.7** | **73.7** | **78.5** | **98.6** | **94.0** | iBOT | **99.2** | **92.2** | **74.6** | **79.6** | **98.9** | **94.3** |

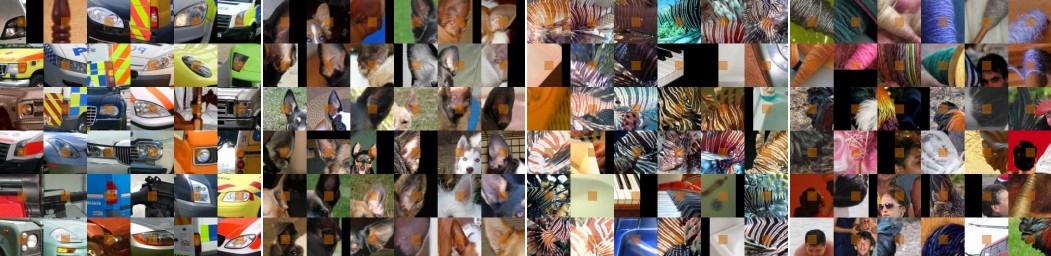

Figure 4: **Pattern layout of patch tokens.** Two left figures showcase patterns, *headlight of the vehicle* and *ear of the dog*, that share part semantics. Two right figures showcase patterns, *stripped* and *curly surface*, that share part textures.

gain over DINO in larger datasets like iNaturalist18 and iNaturalist19, indicating the results are still far from saturation. We also find that with larger models, we typically get larger performance gain compared with DINO (*e.g.*, 1.7% with ViT/S-16 versus 2.0% with ViT-B/16 on iNaturalist18, and 0.3% with ViT/S-16 versus 1.0% with ViT-B/16 on iNaturalist19).

### 4.3 PROPERTIES OF VIT TRAINED WITH MIM

In the previous sections, we have shown the priority of iBOT on various tasks and datasets. To reveal the strengths of iBOT pre-trained Vision Transformers, we analyze its property from several aspects.

#### 4.3.1 DISCOVERING THE PATTERN LAYOUT OF IMAGE PATCHES

**What Patterns Does MIM Learn?** The output from the projection head used for self-distillation depicts for patch token a probabilistic distribution. To help understand what patterns MIM induces to learn, we visualize several pattern layouts. We use 800-epoch pre-trained ViT-S/16 and visualize the top-36 patches with the highest confidence on ImageNet-1K validation set. We visualize a $5\times$ context for each $16 \times 16$ patch (colored orange). We observe the emergence of both high-level semantics and low-level details. As shown in Fig. 4, several patches are grouped with clear semantic meaning, *e.g.*, *headlight* and *dog's ear*. Such behavior stands a distinct contrast with the offline tokenizer used in BEiT (Bao et al., 2021), which encapsulates mostly low-level details as shown in Fig. 16. Apart from patch patterns that share high-level semantics, we also observe clusters accounting for low-level textures, indicating the diversity of learned part patterns. The comparison with previous work (Caron et al., 2021; Bao et al., 2021) and the visualization of more pattern layouts are provided in Appendix G.1.

**How Does MIM Help Image Recognition?** To illustrate how the property of better part semantics can help image recognition, we use *part-wise linear classification* to study the relationship between representations of patch tokens and [CLS] token. Specifically, we average $k$ patch tokens with the top-$k$ highest self-attention scores. The results are demonstrated in Fig. 5. While the performance gap between DINO and iBOT is only 0.9% in the standard setting (77.9% v.s. 77.0%) with [CLS] token, we observe that iBOT outperforms DINO when using the patch representations directly. We observe that using top-56 patch tokens yields an optimal result, and iBOT is 5.9% higher than DINO. The performance gap becomes more prominent when using fewer patch tokens. When using only the patch token with the highest self-attention score, iBOT advances by 17.9%. These results reveal much semantic information in iBOT representations for patch tokens, which helps the model to be more robust to the loss of local details and further boosts its performance on image-level recognition.

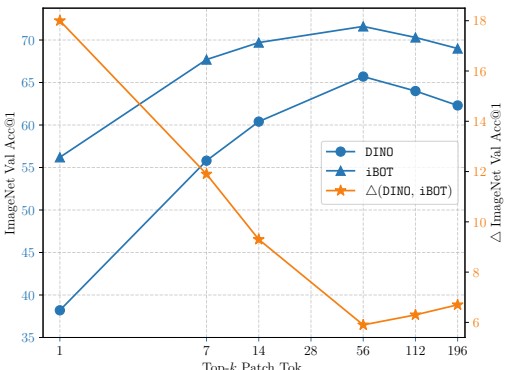

Figure 5: **Part-wise linear probing accuracy.** Top-$k$ tokens with the highest attention scores are averaged for classification.

Figure 6: **Visualization for self-attention map.** Self-attention map from multiple heads are visualized with different color.

Table 8: **Robustness evaluation of pre-trained models against background change, occlusion, and out-of-distribution examples.**

| Method | Background Change | | | | | | | Clean | Occlusion | | Out-of-Dist. | | Clean |
|---|---|---|---|---|---|---|---|---|---|---|---|---|---|
| | *O.F.* | *M.S.* | *M.R.* | *M.N.* | *N.F.* | *O.BB.* | *O.BT.* | IN-9 | $S_{.5}$ | $NS_{.5}$ | IN-A | IN-C$\downarrow$ | IN |
| DINO | 89.2 | 89.2 | 80.4 | 78.3 | 52.0 | 21.9 | **18.4** | 96.4 | 64.7 | 42.0 | 12.3 | 51.7 | 77.0 |
| iBOT | **90.9** | **89.7** | **81.7** | **80.3** | **53.5** | **22.7** | 17.4 | **96.8** | **65.9** | **43.4** | **13.8** | **48.1** | **77.9** |

### 4.3.2 DISCRIMINATIVE PARTS IN SELF-ATTENTION MAP

To analyze, we visualize the self-attention map with ViT-S/16. We choose [CLS] token as the query and visualize attention maps from different heads of the last layer with different colors, as shown in Fig. 6. Of particular interest, we indicate that iBOT shows a solid ability to separate different objects or different parts of one object apart. For example, in the leftmost figure, we observe iBOT fairly distinct the bird from the tree branch. Also, iBOT focuses mainly on the discriminative parts of the object (*e.g.*, *the wheel of the car*, *the beak of the bird*). These properties are crucial for iBOT to excel at image recognition, especially in complicated scenarios with object occlusion or distracting instances. While these properties are not unique strengths brought by MIM and we observe similar behaviors in DINO, we show in Appendix G.2 that iBOT generally gives better visualized results.

### 4.3.3 ROBUSTNESS

The above-mentioned properties brought by MIM objective can improve the model's robustness to uncommon examples. We quantitatively benchmark robustness in terms of 3 aspects: background change, occlusion, and out-of-distribution examples, with a ViT-S/16 pre-trained for 800 epochs and then linearly evaluated for 100 epochs. Results are shown in Tab. 8. For background change, we study images under 7 types of change, detailed in Appendix D. iBOT is more robust against background changes except for *O.BT.*. For occlusion, we study the linear accuracy with salient and non-salient patch dropping following Naseer et al. (2021) with an information loss ratio of 0.5. iBOT has a smaller performance drop under both settings. For out-of-distribution examples, we study natural adversarial examples in ImageNet-A (Hendrycks et al., 2021) and image corruptions in ImageNet-C (Hendrycks & Dietterich, 2019). iBOT has higher accuracy on the ImageNet-A and a smaller mean corruptions error (mCE) on the ImageNet-C.

### 4.4 ABLATION STUDY ON TOKENIZER

In this section, we ablate the importance of using a semantically meaningful tokenizer using a 300-epoch pre-trained ViT-S/16 with a prediction ratio $r = 0.3$ and without multi-crop augmentation. Additional ablations are given in Appendix E. iBOT works with self-distillation on [CLS] token with cross-view images ($\mathcal{L}_{[CLS]}$) to acquire visual semantics. To verify, we conduct experiments to perform MIM without $\mathcal{L}_{[CLS]}$ or with alternative models as visual tokenizer. Specifically, ○ denotes a standalone DINO and △ denotes a pre-tranined DALL-E encoder (Ramesh et al., 2021).

We find that performing MIM without $\mathcal{L}_{\texttt{[CLS]}}$ leads to undesirable results of 9.5% $k$-NN accuracy and 29.8% linear accuracy, indicating that visual semantics can hardly be obtained with only MIM. While semantics emerges with a standalone DINO as a visual tokenizer, it is still far from reaching a decent result (44.3% versus 69.1% in $k$-NN accuracy). Comparing iBOT with multi-tasking of DINO and BEiT (DINO+BEiT), we see the strengths of merging the semantics acquired by self-distillation with the visual tokenizer with an 11.5% advance in linear probing and 0.3% in fine-tuning. More-

Table 9: **Effect of design choices of semantically meaningful tokenization.**

| Method | $\mathcal{L}_{\text{MIM}}$ | $\mathcal{L}_{\texttt{[CLS]}}$ | SH | $k$-NN | Lin. | Fin. |
|---|---|---|---|---|---|---|
| iBOT | ✓ | ✓ | ✓ | 69.1 | 74.2 | 81.5 |
| | ✓ | ✓ | ✗ | 69.0 | 73.8 | 81.5 |
| | ✓ | ✗ | - | 9.5 | 29.8 | 79.4 |
| | ○ | ✗ | - | 44.3 | 60.0 | 81.7 |
| BEiT | △ | ✗ | - | 6.9 | 23.5 | 81.4 |
| DINO | ✗ | ✓ | - | 67.9 | 72.5 | 80.6 |
| BEiT + DINO | △ | ✓ | - | 48.0 | 62.7 | 81.2 |

○: standalone DINO (w/o mcrop, 300-epoch)
△: pre-trained DALL-E encoder

over, we empirically observe a performance improvement using a Shared projection Head (SH) for [CLS] token and patch tokens, which shares the semantics acquired in [CLS] token to MIM.

## 5 RELATED WORK

**Visual Representation Learning.** Most self-supervised methods assume an augmentation invariance of images and achieve so by enforcing similarity over distorted views of one image while avoiding model collapse. Avoiding collapse can be achieved by noise-contrastive estimation with negative samples (Wu et al., 2018; He et al., 2020; Chen et al., 2020a), introducing asymmetric network (Grill et al., 2020; Chen & He, 2021), or explicitly enforcing the distribution of image distribution over the channel to be uniform as well as one-hot (Caron et al., 2020; Amrani & Bronstein, 2021; Caron et al., 2021). In fact, the idea of simultaneously enforcing distribution uniform and one-hot is hidden from earlier studies performing representation learning via clustering (Caron et al., 2018; 2020; YM. et al., 2020), where the cluster assignment naturally meets these two requirements. Other methods rely on handcrafted pretext tasks and assume the image representation should instead be aware of image augmentation by solving image jigsaw puzzle (Noroozi & Favaro, 2016; Wei et al., 2019), predicting rotation (Komodakis & Gidaris, 2018) or relative position (Doersch et al., 2015).

**Masked Prediction in Images.** Predicting masked images parts is a popular self-supervised pretext task drawing on the idea of auto-encoding and has been previously achieved by either recovering raw pixels (Pathak et al., 2016; Atito et al., 2021; Li et al., 2021b) or mask contrastive learning (Henaff, 2020; Zhao et al., 2021). Recently, it is formulated into MIM (Bao et al., 2021; Tan et al., 2021) with a discrete VAE (Rolfe, 2017; Ramesh et al., 2021) as visual tokenizer. As a counterpart of MLM in NLP, MIM eases masked prediction into a classification problem supervised by labels output from the tokenizer, mitigating the problem of excessive focus on high-frequency details. Concurrently, masked image prediction has been explored in the field of multi-modality, *i.e.*, vision-language representation learning. These methods operate on local regions instead of global images thus reply on pre-trained detection models, *i.e.*, Faster-RCNN (Ren et al., 2015) to propose regions of interest. (Su et al., 2020; Lu et al., 2019; Chen et al., 2020c) perform masked region classification tasking the category distribution output from the detection model as the ground-truth.

## 6 CONCLUSION

In this work, we study BERT-like pre-training for Vision Transformers and underline the significance of a semantically meaningful visual tokenizer. We present a self-supervised framework iBOT that performs masked image modeling via self-distillation with an online tokenizer, achieving state-of-the-art results on downstream tasks related to classification, object detection, instance segmentation, and semantic segmentation. Of particular interest, we identify an emerging part-level semantics for models trained with MIM that helps for not only recognition accuracy but also robustness against common image corruptions. In the future, we plan to scale up iBOT to a larger dataset (*e.g.*, ImageNet-22K) or larger model size (*e.g.*, ViT-L/16 and ViT-H/16) and investigate whether MIM can help Vision Transformers more scalable to unlabelled data in the wild.

**Acknowledgement** Tao Kong is the corresponding author. This work was supported by ONR N00014-21-1-2812. Wei Shen was supported by NSFC 62176159 and Natural Science Foundation of Shanghai 21ZR1432200. Cihang Xie was supported by a gift grant from Open Philanthropy. We would like to acknowledge Feng Wang, Rufeng Zhang, and Zongwei Zhou for helpful discussions. We thank Mathilde Caron, Julien Mairal, and Hugo Touvronfor for sharing details of DINO. We thank Li Dong and Hangbo Bao for sharing details of BEiT.

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

## A PSEUDOCODE

---

**Algorithm 1:** iBOT PyTorch-like Pseudocode w/o multi-crop augmentation

---

**Input:**

$g_s, g_t$ ;                                                             // student and teacher network
$C, C'$ ;                                               // center on [CLS] token and patch tokens
$\tau_s, \tau_t$ ;          // temperature on [CLS] token for student and teacher network
$\tau'_s, \tau'_t$ ;       // temperature on patch tokens for student and teacher network
$l$ ;                                                                    // momentum rate for network
$m, m'$ ;           // momentum rates for center on [CLS] token and patch tokens

$g_t$.params = $g_s$.params

**for** $x$ in loader **do**

    $u, v$ = augment($x$), augment($x$) ;                                         // random views

    $\hat{u}, m_u$ = blockwise_mask($u$) ;                      // random block-wise masking
    $\hat{v}, m_v$ = blockwise_mask($v$) ;                      // random block-wise masking

    $\hat{u}_s^{\text{[CLS]}}, \hat{u}_s^{\text{patch}} = g_s(\hat{u}, \text{return\_all\_tok=true})$ ;                    // $[n, K]$, $[n, S^2, K]$
    $\hat{v}_s^{\text{[CLS]}}, \hat{v}_s^{\text{patch}} = g_s(\hat{v}, \text{return\_all\_tok=true})$ ;                    // $[n, K]$, $[n, S^2, K]$

    $u_t^{\text{[CLS]}}, u_t^{\text{patch}} = g_t(u, \text{return\_all\_tok=true})$ ;                    // $[n, K]$, $[n, S^2, K]$
    $v_t^{\text{[CLS]}}, v_t^{\text{patch}} = g_t(v, \text{return\_all\_tok=true})$ ;                    // $[n, K]$, $[n, S^2, K]$

    $\mathcal{L}_{\text{[CLS]}} = \text{H}(\hat{u}_s^{\text{[CLS]}}, v_t^{\text{[CLS]}}, C, \tau_s, \tau_t) / 2 + \text{H}(\hat{v}_s^{\text{[CLS]}}, u_t^{\text{[CLS]}}, C, \tau_s, \tau_t) / 2$
    $\mathcal{L}_{\text{MIM}} = (m_u \cdot \text{H}(\hat{u}_s^{\text{patch}}, u_t^{\text{patch}}, C', \tau'_s, \tau'_t)\text{.sum(dim=1)} / m_u\text{.sum(dim=1)} / 2$
        $+ (m_v \cdot \text{H}(\hat{v}_s^{\text{patch}}, v_t^{\text{patch}}, C', \tau'_s, \tau'_t)\text{.sum(dim=1)} / m_v\text{.sum(dim=1)} / 2$
    $(\mathcal{L}_{\text{[CLS]}}\text{.mean()} + \mathcal{L}_{\text{MIM}}\text{.mean())}\text{.backward()}$

    update($g_s$) ;                                    // student, teacher and center update
    $g_t$.params = $l \cdot g_t$.params $+ (1 - l) \cdot g_s$.params
    $C = m \cdot C + (1 - m) \cdot \text{cat}([u_t^{\text{[CLS]}}, v_t^{\text{[CLS]}}])\text{.mean(dim=0)}$
    $C' = m' \cdot C' + (1 - m') \cdot \text{cat}([u_t^{\text{patch}}, v_t^{\text{patch}}])\text{.mean(dim=(0, 1))}$
**end**

**def** H ($s$, $t$, $c$, $\tau_s$, $\tau_t$) **:**
    $t = t$.detach();                                                      // stop gradient
    $s = \text{softmax}(s / \tau_s, \text{dim=1})$
    $t = \text{softmax}((t - c) / \tau_t, \text{dim=1})$;                        // center + sharpen
    **return** $-(t \cdot \log(s))\text{.sum(dim=-1)}$;

---

## B MULTI-CROP

The advanced performance of several recent state-of-the-art methods (Caron et al., 2021; 2020) relies on multi-crop augmentation, as well as iBOT. In our early experiments, we find the direct usage of multi-crop augmentation leads to instability issues that degrade accuracy. We reveal that these results can be attributed to the distribution mismatch between masked images and non-masked images and can be resolved by minimal changes in iBOT framework.

**Stability of MIM Pre-trained with Multi-Crop.** We first showcase several practices where training instability occurs, shown in Fig. 7. To reveal the instability, we monitor the NMI curves during training for each epoch as shown in Fig. 8. The most intuitive ideas are to compute as (b) or (c). In (b), MIM is only performed on global crops. This pipeline is unstable during training, and we observe a dip in the NMI training curve. We hypothesize that it can be caused by the distribution mismatch of masked global crops and non-masked local crops. To alleviate this, a straightforward solution is to also perform MIM on local crops with an extra computation cost as (c). However, we do not observe this circumvents training instability. We hypothesize that the regions corresponding

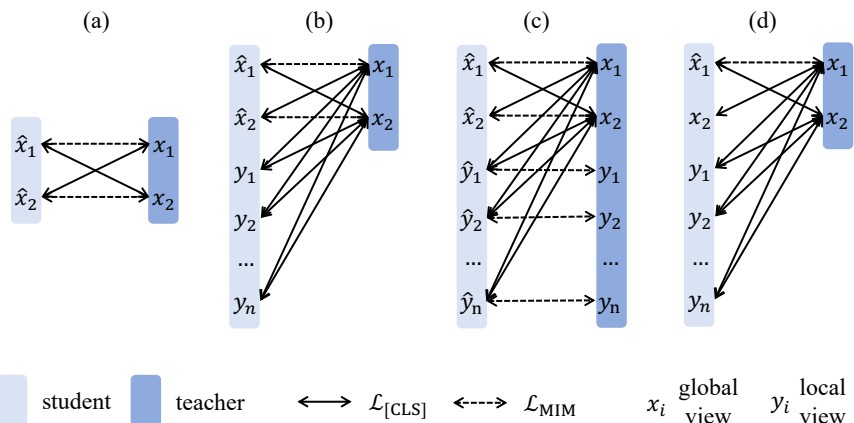

Figure 7: **Computation pipelines for iBOT with or without multi-crop augmentation.** (a) iBOT w/o multi-crop augmentation. (b), (c), and (d) are three pipelines w/ multi-crop augmentation. (b) does not perform MIM for local crops, whereas (c) performs MIM for all crops. (d) only performs MIM for one of the two global crops. iBOT uses (b) with random MIM.

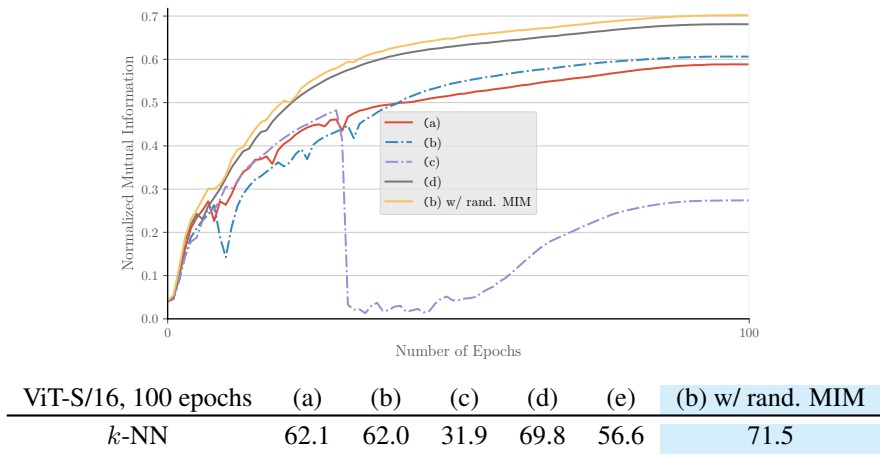

| ViT-S/16, 100 epochs | (a) | (b) | (c) | (d) | (e) | (b) w/ rand. MIM |
|---|---|---|---|---|---|---|
| $k$-NN | 62.1 | 62.0 | 31.9 | 69.8 | 56.6 | 71.5 |

Figure 8: **Training curves of different multi-crop strategy.**

to patch tokens of the local crops are small in size, in which there exist few meaningful contents to predict. This hypothesis can be supported by the experiments that when we set the local crop scale in (c) from $(0.05, 0.4)$ to $(0.2, 0.4)$, denoted as (e), the performance drop is mitigated.

**Stabilizing the Training with Non-Masked Global Crops.** Another solution to alleviate the distribution mismatch between masked global crops and non-masked local crops is to train with non-masked global crops, as shown in (d). In other words, we perform ***random MIM*** when training ViT with multi-crop augmentation. This computation pipeline is stable and achieves a substantial performance gain. In practice, to include non-masked global crops in training, we use (b) and randomly choose a prediction ratio between $[0, r \ (r > 0)]$ for each image. When the ratio $0$ is chosen, the whole framework excludes MIM and can be seen as DINO. When the ratio $r \ (r > 0)$ is chosen, MIM is performed for both of the two global crops. We observe the latter practice performs sightly better since it is more flexible in task composition and data in a batch is mutually independent.

**Range of Scales in Multi-Crop.** We further study the performance with different local and global scale. Following DINO (Caron et al., 2021), we conduct the experiments by tweaking $s$, where $s$ is the scale deviding the local and global crops. The local crops are sampled from $(0.05, s)$ and the global crops are sampled from $(s, 1)$.

| ViT-S/16, 300 epochs | 0.25 | 0.4 | 0.32 | | ViT-B/16, 50 epochs | 0.25 | 0.4 | 0.32 |
|---|---|---|---|---|---|---|---|---|
| $k$-NN | 74 | 74.3 | 74.6 | | $k$-NN | 70 | 70.1 | 70.4 |

Table 10: $k$-**NN and linear probing accuracy on ImageNet-1K without multi-crop augmentation (left) and with multi-crop augmentation (right) multi-crop augmentation**. We split the table into results without or with multi-crop augmentation.

| Method | Arch | Param. | Epo. | $k$-NN | Linear | Method | Arch | Param. | Epo. | $k$-NN | Linear |
|---|---|---|---|---|---|---|---|---|---|---|---|
| MoCov3 | RN50 | 23 | 800 | - | 74.6 | SwAV | RN50 | 23 | 800 | 65.7 | 75.3 |
| | ViT-S/16 | 21 | 600 | - | 73.4 | | ViT-S/16 | 21 | 800 | 66.3 | 73.5 |
| | ViT-B/16 | 85 | 600 | - | 76.7 | | | | | | |
| DINO | ViT-S/16 | 21 | 800 | 70.0 | 73.7 | DINO | RN50 | 23 | 800 | 67.5 | 75.3 |
| | ViT-B/16 | 85 | 400 | 68.9 | 72.8 | | ViT-S/16 | 21 | 800 | 74.5 | 77.0 |
| | | | | | | | ViT-B/16 | 85 | 400 | 76.1 | 78.2 |
| iBOT | ViT-S/16 | 21 | 800 | **72.4** | **76.2** | iBOT | ViT-S/16 | 21 | 800 | **75.2** | **77.9** |
| | ViT-B/16 | 85 | 400 | **71.2** | **76.0** | | ViT-B/16 | 85 | 400 | **76.8** | **79.4** |

We empirically find that $s = 0.32$ yields optimal performance for both small-size and base-size models. Therefore, we use an $s$ of $0.32$ by default.

**State-of-the-Art Comparison w/o and w/ Multi-Crop.** Including iBOT, several recent state-of-the-art works (Caron et al., 2021; 2020) rely heavily on multi-crop augmentation during pre-training. Except for several specific self-supervised methods (Grill et al., 2020), multi-crop works well on most of the self-supervised methods and consistently yields performance gain (Caron et al., 2021). While a more fair comparison with our methods without multi-crop augmentation can be conducted, we believe it is a unique strength of iBOT to work well with multi-crop. In Tab. 10, we categorize the state-of-the-art comparison into two parts where one for methods without multi-crop and the other with multi-crop. For the former, we mainly compare our method without multi-crop with MoCov3 (Chen et al., 2021) and DINO without multi-crop. We observe that our method achieves state-of-the-art performance with ViT-S/16 even without multi-crop and comparable performance with ViT-B/16 compared with MoCov3. For the latter, we mainly compare our method with SwAV (Caron et al., 2020) and DINO with multi-crop augmentation. We observe that iBOT achieves higher performance with 79.4% of linear probing accuracy when using ViT-S/16.

**Effective Training Epochs.** Due to extra computation costs brought by multi-crop augmentation, different methods with the same pre-training epochs actually see different total numbers of images. To mitigate, we propose to measure the effective training epochs, defined as actual pre-training epochs multiplied with a scaling factor accounting for extra trained images of different resolutions induced by multi-crop augmentation. DINO and iBOT are by default trained with 2 global crops of size $224 \times 224$ and 10 local crops of size $96 \times 96$. Thus $r = 2 + (\frac{96}{224})^2 \times 10 = 3.84 \approx 4$ for DINO and iBOT. $r \approx 3$ for SwAV or DINO with RN50 as the backbone and pre-trained with 2 global crops and 6 local crops. $r = 2$ for contrastive methods without multi-crop augmentation (*e.g.*, MoCo, SimCLR, BYOL, *etc.*) and $r = 1$ for non-contrastive methods (*e.g.*, BEiT, Jigsaw, *etc.*).

## C    ADDITIONAL IMPLEMENTATIONS

**Fine-Tuning Recipes of Classification on ImageNet-1K.** By default, we follow the fine-tuning protocol in BEiT (Bao et al., 2021) to use a layer-wise learning rate decay, weight decay and AdamW optimizer and train small-, base-size models with 200, 100, and 50 epochs respectively. We sweep over four learning rates $\{8e^{-4}, 9e^{-4}, 1e^{-3}, 2e^{-3}\}$. Comparatively, traditional fine-tuning recipe is is to fine-tune the network for 300 epochs with a learning rate $5e^{-4}$, no weight decay, and SGD optimizer (Touvron et al., 2021) (Row 1 versus 8). For a fair comparison, we compare the impact of different fine-tuning recipes with different methods, shown in Tab. 11. We empirically find that fine-tuning protocol used in BEiT consistently yields better fine-tuning results and greatly reduces the training epochs. By default, we use a layerwise decay of $0.75$ with a training epoch of 200 for ViT-S/16, a layerwise decay of $0.65$ with a training epoch of 100 for ViT-B/16, and a layerwise decay of $0.75$ with a training epoch of 50 for ViT-L/16. We report the higher results between using or not using DS since we find it brings different impacts to different methods.

**Evaluation Protocols of Semi-Supervised Learning on ImageNet-1K.** We study the impact of different evaluation protocols for semi-supervised learning. Under conventional semi-supervised evaluation protocol, pre-trained models are end-to-end fine-tuned with a linear classification head.

Table 11: **Different fine-tuning recipes.** LD denotes layerwise learning rate decay. DS denotes mixed-precision training with DeepSpeed.

|   | Epo. | LD | DS | BEiT | DINO | iBOT |
|---|------|-----|-----|------|------|------|
| *ViT-S/16* | | | | | | |
| 1 | 300 | 1.0 | ✗ | 81.5 | 81.1 | 81.2 |
| 2 | 300 | 0.75 | ✓ | **81.7** | 82.0 | 82.3 |
| 3 | 200 | 0.65 | ✗ | 80.7 | - | - |
| 4 | 200 | 0.75 | ✗ | 81.4 | 81.9 | **82.3** |
| 5 | 200 | 0.75 | ✓ | 81.4 | **82.0** | 82.2 |
| 6 | 200 | 0.85 | ✗ | 81.2 | - | - |
| *ViT-B/16* | | | | | | |
| 7 | 300 | 1.0 | ✗ | 82.1 | 82.8 | 82.4 |
| 8 | 200 | 0.65 | ✓ | 82.7 | 83.1 | 83.2 |
| 9 | 100 | 0.65 | ✗ | **83.4** | 83.5 | **84.0** |
| 10 | 100 | 0.65 | ✓ | 83.2 | **83.6** | 83.8 |

Table 12: **Evaluation protocols for semi-supervised learning.** *Proj.* denotes fine-tuning from the middle layer of the projection head. LR denotes logistic regression.

|   | Method | | *Proj.* | 1% | 10% |
|---|--------|-----|------|------|------|
| *frozen features* | | | | | |
| 1 | DINO | + $k$-NN | - | 61.3 | 69.1 |
| 2 | iBOT | + $k$-NN | - | 62.3 | 70.1 |
| 3 | DINO | + Lin. | - | 60.5 | 71.0 |
| 4 | iBOT | + Lin. | - | 62.5 | 72.2 |
| 5 | DINO | + LR | - | 64.5 | 72.2 |
| 6 | iBOT | + LR | - | **65.9** | 73.4 |
| *end-to-end fine-tuning* | | | | | |
| 7 | DINO | | ✗ | 50.6 | 73.2 |
| 8 | iBOT | | ✗ | 55.0 | 74.0 |
| 9 | DINO | | ✓ | 60.3 | 74.3 |
| 10 | iBOT | | ✓ | 61.9 | **75.1** |

SimCLRv2 Chen et al. (2020b) found that keeping the first layer of the projection head can improve accuracy, especially under the low-shot setting. We fine-tune the pre-trained model from the first layer of the projection head and verify this conclusion holds true for Vision Transformers. We empirically find that Vision Transformer performs better with a frozen backbone with 1% of training data (62.5% in row 4 versus 61.9 % in row 7). In DINO, a logistic regressor built upon the frozen features is found to perform better compared with the multi-class linear classifier upon the frozen features, especially with 1% data (65.9% in row 6 versus 62.5% in row 4). When using 10% data, we empirically find that *end-to-end fine-tuning* from the first layer of the projection layer yields the best performance (75.1% in row 10 versus 73.4% in row 6).

**Fine-Tuning Recipes of Object Detection and Instance Segmentation on COCO.** For both small- and base-size models, we utilize multi-scale training (resizing image with shorter size between 480 and 800 while the longer side no larger than 1333), a learning rate $1e^{-4}$, a weight decay of 0.05, and fine-tune the entire network for $1\times$ schedule (12 epochs with the learning rate decayed by $10\times$ at epochs 9 and 11). We sweep a layer decay rate of $\{0.65, 0.75, 0.8, 0.9\}$. Note that a layer decay rate of 1.0 denotes no layer is decayed. To produce hierarchical feature maps, we use the features output from layer 4, 6, 8, and 12, with 2 deconvolutions, 1 deconvolution, identity mapping, and max-pooling appended after, respectively. We do not use multi-scale testing.

**Fine-Tuning Recipes of Semantic Segmentation on ADE20K.** For semantic segmentation, we follow the configurations in BEiT (Bao et al., 2021), fine-tuning 160k iterations with $512 \times 512$ images and a layer decay rate of 0.65. We do not use multi-scale training and testing. We sweep the learning rate $\{3e^{-5}, 8e^{-5}, 1e^{-4}, 3e^{-4}, 8e^{-4}\}$. Similar to object detection and instance segmentation, to produce hierarchical feature maps, we add additional deconvolution layers after ViT.

| DINO, w/o `[LN]` | DINO, w/ `[LN]` | iBOT, w/o `[LN]` | iBOT, w/ `[LN]` |
|------|------|------|------|
| 33.7 | **34.5** | 37.8 | **38.3** |

When using linear (Lin.) as the task layer, we find that appending the last LayerNorm (`[LN]`) for `[CLS]` token to each patch tokens before the decoder consistently yields better performance, while we do not spot the substantial gain when with UperNet as the task layer. By default, we report the segmentation result with `[LN]` for both linear head for UperNet head.

**Part-Wise Linear Probing.** We use the average of the last-layer self-attention map with `[CLS]` as the query from multiple heads to rank all the patch tokens. We remove the extra LayerNorm (LN) after the final block following MoCov3 (Chen et al., 2021).

# D ADDITIONAL RESULTS

In this section, we provide detailed results for dense downstream tasks, *i.e.*, object detection, instance segmentation, and semantic segmentation. We give the complete figures for occlusion robustness

Table 13: **Additional object detection, instance segmentation, and semantic segmentation results with small-size models.** We pre-train iBOT with ViT-S/16 for 800 epochs.

| Method | Arch. | Param. | Det. & Inst. Seg. w/ Cascade Mask R-CNN | | | | | | Seg. w/ UperNet | |
|--------|-------|--------|-----------|-----------|-----------|-----------|-----------|-----------|-------|------|
| | | | $AP^b$ | $AP^b_{50}$ | $AP^b_{75}$ | $AP^m$ | $AP^m_{50}$ | $AP^m_{75}$ | mIoU | mAcc |
| Sup. | Swin-T | 29 | 48.1 | 67.1 | 52.5 | 41.7 | 64.4 | 45.0 | 44.5 | - |
| MoBY | Swin-T | 29 | 48.1 | 67.1 | 52.1 | 41.5 | 64.0 | 44.7 | 44.1 | - |
| Sup. | ViT-S/16 | 21 | 46.2 | 65.9 | 49.6 | 40.1 | 62.9 | 42.8 | 44.5 | 55.5 |
| iBOT | ViT-S/16 | 21 | **49.4** | **68.7** | **53.3** | **42.6** | **65.6** | **45.8** | **45.4** | **56.2** |

Table 14: **Additional object detection, instance segmentation, and semantic segmentation results with base-size models.** We pre-train iBOT with ViT-B/16 for 400 epochs.

| Method | Det. & Inst. Seg. w/ Cascade Mask R-CNN | | | | | | Seg. w/ Lin. | | Seg. w/ UperNet | |
|--------|-----------|-----------|-----------|-----------|-----------|-----------|-------|------|-------|------|
| | $AP^b$ | $AP^b_{50}$ | $AP^b_{75}$ | $AP^m$ | $AP^m_{50}$ | $AP^m_{75}$ | mIoU | mAcc | mIoU | mAcc |
| Sup. | 49.8 | 69.6 | 53.8 | 43.2 | 66.6 | 46.5 | 35.4 | 44.6 | 46.6 | 57.0 |
| BEiT | 50.1 | 68.5 | 54.6 | 43.5 | 66.2 | 47.1 | 27.4 | 35.5 | 45.8 | 55.9 |
| DINO | 50.1 | 69.5 | 54.3 | 43.4 | 66.8 | 47.0 | 34.5 | 43.7 | 46.8 | 57.1 |
| iBOT | **51.2** | **70.8** | **55.5** | **44.2** | **67.8** | **47.7** | **38.3** | **48.0** | **50.0** | **60.3** |

Table 15: $k$-**NN and linear probing on ImageNet-1K with different pre-training datasets.**

| Arch. | Pre-Train Data | Param. | Epoch | $k$-NN | Linear |
|-------|----------------|--------|-------|--------|--------|
| ViT-S/16 | ImageNet-1K | 21 | 800 | 75.2 | 77.9 |
| ViT-S/16 | ImageNet-22K | 21 | 160 | 69.3 | 76.5 |
| ViT-B/16 | ImageNet-1K | 85 | 400 | 77.1 | 79.5 |
| ViT-B/16 | ImageNet-22K | 85 | 80 | 71.1 | 79.0 |
| ViT-L/16 | ImageNet-1K | 307 | 300 | **78.0** | 81.0 |
| ViT-L/16 | ImageNet-22K | 307 | 50 | 72.9 | **82.3** |

Table 16: **Effectiveness of pre-trained features on nearest neighbor retrieval.** We report the results on different downstream tasks whose evaluation is based on nearest neighbor retrieval.

| Method | Image Retrieval | | | | Vid. Obj. Segment. | | |
|--------|------|------|------|------|------------------|------|------|
| | $\mathcal{R}$Ox | | $\mathcal{R}$Par | | | | |
| | M | H | M | H | $(\mathcal{J}\&\mathcal{F})_m$ | $\mathcal{J}_m$ | $\mathcal{F}_m$ |
| DINO | **37.2** | **13.9** | **63.1** | **34.4** | 61.8 | 60.2 | **63.4** |
| iBOT | 36.6 | 13.0 | 61.5 | 34.1 | **61.8** | **60.4** | 63.2 |

analysis. We also provide extra experiments of nearest neighbor retrieval, robustness analysis against occlusion and shuffle.

**Object Detection, Instance Segmentation, and Semantic Segmentation.** We here provide more detailed results on object detection, instance segmentation, and semantic segmentation with small- and base-size models, shown in Tab. 13 and Tab. 14 respectively. Specifically, we include $AP^b_{50}$ and $AP^b_{75}$ for object detection, $AP^m_{50}$ and $AP^m_{75}$ for instance segmentation, mAcc for semantic segmentation. For object detection (Det.) and instance segmentation (Inst. Seg.), we consider Cascade Mask R-CNN as the task layer. For semantic segmentation (Seg.), we consider two evaluation settings where a linear head (Lin.) and UPerNet are taken as the task layer.

$k$-**NN and Linear Probing with ImageNet-22K.** We further report $k$-NN and linear probing accuracy on ImageNet-1K with models pre-trained on ImageNet-22K dataset. We empirically observe that ImageNet-1K pre-training incurs better ImageNet-1K $k$-NN and linear probing performance, which is opposite to the fine-tuning performance observed in Table 2 and Table 3. We hypothesize that the data distribution plays a more crucial rule under evaluation protocols based on frozen features, such that models pre-trained with smaller ImageNet-1K dataset consistently achieve better results.

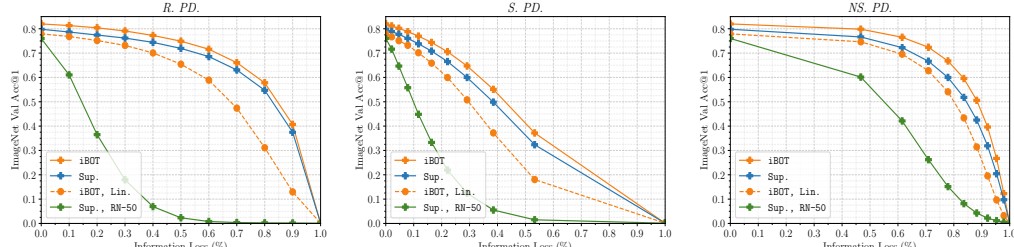

Figure 9: **Robustness against occlusion.** Model's robustness against occlusion with different information loss ratios is studied. 3 patch dropping settings: Random Patch Dropping (left), Salient Patch Dropping (middle), and Non-Salient Patch Dropping (right) are considered.

**Nearest Neighbor Retrieval.** Nearest neighbor retrieval is considered using the frozen pre-trained features following the evaluation protocol as in DINO (Caron et al., 2021). DINO has demonstrated the strong potential of pre-trained ViT features to be directly used for retrieval. To validate, DINO designed several downstream tasks, including image retrieval and video object segmentation, where video object segmentation can be seen as a dense retrieval task by finding the nearest neighbor between consecutive frames to propagate masks. We compare iBOT with DINO on these benchmarks with the same evaluation settings. As demonstrated in Tab. 16, iBOT has comparable results with DINO. While iBOT has higher $k$-NN results on Imagenet-1K, the performance is not better for iBOT in image retrieval. We empirically find that the results on image retrieval are sensitive to image resolution, multi-scale features, *etc.*, and the performance varies using pre-trained models with minimal differences on hyper-parameter setup. For this reason, we do not further push iBOT for better results.

**Robustness against Background Change.** Deep models rely on both foreground objects and backgrounds. Robust models should be tolerant to background changes and able to locate discriminative foreground parts. We evaluate this property on ImageNet-9 (IN-9) dataset (Xiao et al., 2020). IN-9 includes 9 coarse-grained classes and 7 variants by mixing up the foreground and background from different images. *Only-FG (O.F.)*, *Mixed-Same (M.S.)*, *Mixed-Rand (M.R.)*, and *Mixed-Next (M.N.)* are 4 variant datasets where the original foreground is present but the background is modified, whereas *No-FG (N.F.)*, *Only-BG-B (O.BB.)*, and *Only-BG-T (O.BT.)* are 3 variants where the foreground is masked. As shown in Tab. 8, we observe a performance gain except for *O.BT.*, indicating iBOT's robustness against background changes. We note in *O.BT.* neither foreground nor foreground mask is visible, contradicting the pre-training objective of MIM.

**Robustness against Occlusion.** Masked prediction has a natural strength in cases where parts of the image are masked out since the models are trained to predict their original contents. We here provide the detailed results of occlusion with different information loss ratios in Fig. 9 under three dropping settings: random, salient, and non-salient. We showcase the results of iBOT end-to-end fine-tuned or with a linear head over the pre-trained backbone. We include the results of supervised results with both ViT-S/16 and ResNet-50 for comparison. ViT shows higher robustness compared to its CNN counterpart, *i.e.*, ResNet-50, given that Transformers' dynamic receptive field makes it less dependent on images' spatial structure. We empirically find iBOT has stronger robustness against occlusion compared to its supervised baseline, consolidating that MIM help to model the interaction between the sequence of image patches using self-attention such that discarding proportion of elements does not degrade the performance significantly.

**Robustness against Shuffle.** We study the model's sensitivity to the spatial structure by shuffling on input image patches. Specifically, we shuffle the image patches with different grid sizes following (Naseer et al., 2021). We showcase the results of iBOT end-to-end fine-tuned or with a linear head over the pre-trained backbone. We include the results of supervised results with both ViT-S/16 and ResNet-50 for comparison. Note that a shuffle grid size of 1 means no shuffle, and a shuffle grid size of 196 means all patch tokens are shuffled. Fig. 10 suggests that iBOT retain accuracy better than its supervised baseline and ResNet-50. It also indicates that iBOT relies less on positional embedding to preserve the global image context for right classification decisions.

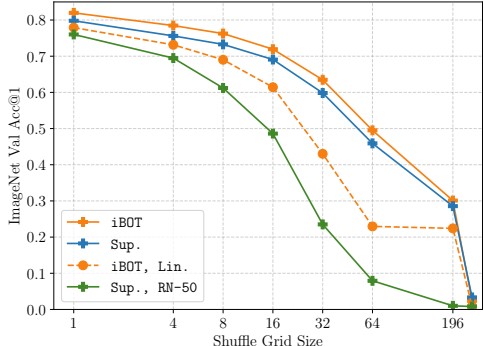

Figure 10: **Robustness against shuffle.** Model's robustness against shuffle with different grid shuffle sizes is studied.

# E    ADDITIONAL ABLATIONS

In this section, we study the impact of other parameters that we have conducted experiments on. Without extra illustrations, we use 300-epoch pre-trained ViT-S/16, a prediction ratio $r = 0.3$ and without multi-crop augmentation for the ablative study.

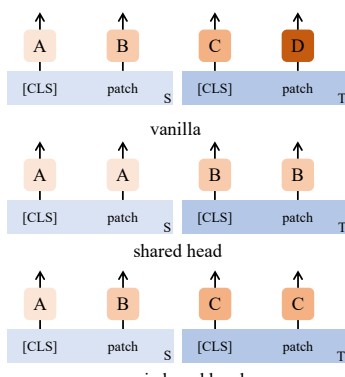

Table 17: **Different head sharing strategy.**

Table 18: **Hard label versus soft label.** *Cen.* denotes centering. † denotes smaller temperature for teacher output.

| Method | *Cen.* | Post Proc. | $k$-NN | Lin. |
|---|---|---|---|---|
| | ✗ | softmax | 49.8 | 63.5 |
| | ✗ | hardmax | 64.8 | 71.9 |
| | ✗ | softmax† | 69.4 | 73.9 |
| | ✓ | softmax | 67.8 | 72.9 |
| | ✓ | hardmax | 68.1 | 73.3 |
| iBOT | ✓ | softmax† | 69.1 | 74.2 |
| DINO | - | - | 67.9 | 72.5 |

**Architecture of Projection Head.** As mentioned earlier, a shared head can transfer the semantics acquired in [CLS] token to patch tokens, slightly improving the performance. We notice that the head for patch tokens in the student network only see the masked tokens throughout the training, the distribution of which mismatches tokens with natural textures. Therefore, we conduct an experiment using a non-shared head for the student network but a shared head for the teacher network denoted as *semi-shared head*. Their differences are demonstrated in Fig. 17, where $S$ and $T$ denotes student and teacher network respectively. The heads with the same index and color denotes they have shared parameters.

| Arch. | vanilla | shared† | sm. shared | sm. shared† | shared |
|---|---|---|---|---|---|
| $k$-NN . | 68.9 | 68.0 | 68.4 | 68.4 | 69.1 |
| Lin. | 73.9 | 73.7 | 73.7 | 73.8 | 74.2 |

† denotes only the first 2 layers out of the 3-layer MLP share the parameters. However, we do not observe that semi-shared head is better than shared head. By default, we share the entire projection head for [CLS] token and patch tokens.

**Comparing MIM with Dense Self-Distillation.** To identify the superiority of MIM to model internal structure using over its alternatives, we conduct experiments performing self-distillation on original patch tokens along with the [CLS] token. We consider two matching strategies to construct patch token pairs for self-distillation.

| Arch. | DINO | DINO + *pos.* | DINO + *feat.* | iBOT |
|---|---|---|---|---|
| $k$-NN | 67.9 | 67.1 ($-0.8$) | 68.5 ($+0.6$) | 69.1 ($+1.2$) |
| Lin. | 72.5 | 72.5 ($+0.0$) | 73.4 ($+0.9$) | 74.2 ($+1.7$) |

Specifically, *pos.* denotes matching according to the absolute position of two views. Similar to Xie et al. (2021b). $j$ is defined as $\arg\min_j dist(p_i, p'_j)$, where $p$ is the position in the original image space and $dist(u, v)$ is euclidean distance. The losses are only computed for the overlapped regions of two views. We do not observe substantial gain brought by matching via patches' absolute position. *feat.* denotes matching according to the similarity of the backbone similarity of two views. Similar to Wang et al. (2021b), we match for each patch token $f_i$ the most similar patch token from another view $f'_j$, where $j = \arg\max_j sim(f_i, f'_j)$. $sim(u, v)$ is cosine distance. Such practice brings a $0.6\%$ performance gain in terms of linear probing accuracy, which is also observed by a concurrent work, EsViT (Li et al., 2021a). Comparatively, iBOT prompts an $1.2\%$ gain on linear probing, verifying the necessity and advancement of MIM.

**Hard Label versus Soft Label** We study the importance of using a continuous token distribution (softmax[†]) instead of a discretized id (hardmax) when performing MIM. Results in Tab. 18 indicate continuous tokenization plays a crucial part. We empirically find the improvement brought by centering, whose roles are less important compared to centering in self-distillation on `[CLS]` token. Only sharpening can produce a $k$-NN accuracy of 69.4 and a linear probing accuracy of 73.9.

**Centering and Sharpening.** Different from the `[CLS]` token, patch tokens do not have certain semantic cluster and vary more widely from each others. We study the impact of several critical parameters that decide the distillation process and customize them for distillation over the patch tokens.

| $m'$ | .8 | .99 | .999 | .9 | .9 | .9 |
|---|---|---|---|---|---|---|
| $\tau'_t$ | $.04 \to .07$ | $.04 \to .07$ | $.04 \to .07$ | $.04 \to .06$ | $.05 \to .08$ | $.04 \to .07$ |
| $k$-NN | 68.7 | 68.8 | 68.9 | 68.5 | 68.7 | 69.1 |
| Lin. | 74.0 | 73.8 | 73.8 | 73.5 | 73.9 | 74.2 |

Specifically, the smoothing momentum for online centering $m'$ and sharpening temperature $\tau'_t$ are studied. Note we keep the parameters for `[CLS]` token the same as DINO and only study for parameters for the patch tokens.

**Loss Ratio.** We study the impact of different ratio between $\mathcal{L}_{\texttt{[CLS]}}$ and $\mathcal{L}_{\text{MIM}}$. We keep the base of $\mathcal{L}_{\texttt{[CLS]}}$ to 1 and scale $\mathcal{L}_{\text{MIM}}$ with different ratios.

| $\mathcal{L}_{\texttt{[CLS]}}$ / $\mathcal{L}_{\text{MIM}}$ | 0.5 | 2 | 1 |
|---|---|---|---|
| $k$-NN | | 68.7 | 69.4 | 69.1 |
| Lin. | | 73.8 | 74.1 | 74.2 |

We observe that directly adding two losses up without scaling yields the best performance in terms of linear probing accuracy.

**Output Dimension.** We follow the structure of projection head in DINO with l2-normalized bottleneck and without batch normalization. We study the impact of output dimension $K$ of the last layer.

| $K$ | 4096 | 16384 | 8192 |
|---|---|---|---|
| $k$-NN | 68.3 | 68.8 | 69.1 |
| Lin. | 74.5 | 74.0 | 74.2 |

While our method excludes large output dimensionality since each patch token has an output distribution, we do not observe substantial performance gain brought by larger output dimensions. Therefore, we choose $K = 8192$ by default.

**Prediction Ratios.** Masked modeling is based on a formulation of partial prediction, the objective of which is to maximize the log-likelihood of the target tokens conditioned on the non-target tokens. We experiment with different prediction ratios for masked image modeling. The results are shown in Fig. 11. We observe that the performance is not sensitive to variant prediction ratios between $0.05$ and $0.4$. Adding a variance upon the fixed value can also consistently bring a performance gain, which can be explained as stronger data augmentation. The teacher output of non-masked images is now pulled together with the student output of masked images with different ratios. By default, we use $0.3$ ($\pm 0.2$) as the prediction ratio. For models with multi-crop augmentation, following the above discussions, we randomly choose a prediction of 0 or $0.3$ ($\pm 0.2$) for each image.

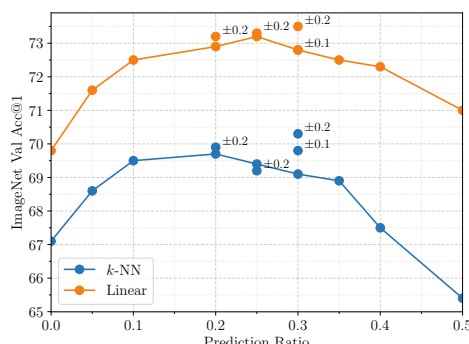

Figure 11: **Impact of the prediction ratio.**
$\pm$ denotes to randomly sample from a region.

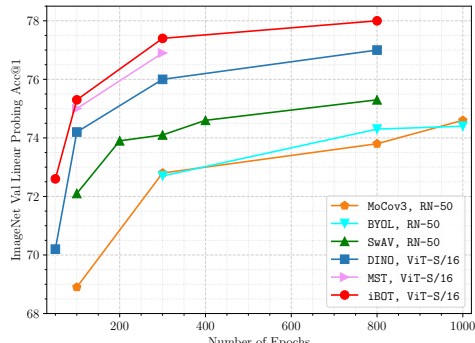

Figure 12: **Impact of the training epochs.**
Models are ViT-S/16 with multi-crop augmentation.

Table 19: **Time and Memory Requirements.** We detail the actual training time (T) and GPU memory (Mem.) of different methods, together with their respective linear probing (Lin.) and fine-tuning (Fin.) accuracy. All methods are trained on two 8-GPU V100 machines with a batch size of 1024.

| Method | Crops Number | $T_{100}$ | $T_{300}$ | $T_{800}$ | Mem. | $Lin._{300}$ | $Lin._{800}$ | $Fin._{800}$ |
|---|---|---|---|---|---|---|---|---|
| BEiT | $1 \times 224^2$ | 11.3h | 33.7h | 90.1h | 5.6G | 20.7 | 24.2 | 81.4 |
| DINO | $2 \times 224^2$ | 15.1h | 44.7h | 111.6h | 9.3G | 72.5 | 73.7 | 81.6 |
| iBOT | $2 \times 224^2$ | 15.6h | 47.0h | 126.4h | 13.1G | 74.8 | 76.2 | 82.0 |
| DINO | $2 \times 224^2 + 10 \times 96^2$ | 24.2h | 72.6h | 180.0h | 15.4G | 76.2 | 77.0 | 82.0 |
| iBOT | $2 \times 224^2 + 10 \times 96^2$ | 24.3h | 73.3h | 193.4h | 19.5G | 77.4 | 77.9 | 82.3 |

Table 20: **Methodology comparison over different approaches to tokenize the patches.** We report ImageNet-1K $k$-NN, linear and fine-tuning validation accuracy. Models are pre-trained with ViT-S/16 and 300 epochs.

| Method | $k$-NN | Linear | Fine-Tune |
|---|---|---|---|
| Rand. | - | - | 79.9 |
| MPP (Dosovitskiy et al., 2021) | 16.4 | 37.2 | 80.8 |
| Patch Clustering | 19.2 | 40.1 | 81.3 |
| BEiT (Bao et al., 2021) | 6.9 | 24.2 | 81.4 |
| Standalone DINO as tokenizer | 44.3 | 60.0 | 81.7 |
| iBOT | 70.3 | 74.8 | 81.5 |

**Training Epochs.** We provide the linear probing top-1 accuracy with ViT-S/16 pre-trained for different epochs. For comparison, we also include the accuracy curve of other methods with comparable numbers of parameters, *i.e.*, ResNet-50. From Fig. 12, we observe that longer training for 800 epochs can improve the model's performance. It's north worthy that iBOT can achieve a Top-1 accuracy of SwAV (Caron et al., 2020) pre-trained with 800 epochs in less than 100 epochs. iBOT pre-trained with 800 epochs brings a 0.9% improvement over previous state-of-the-art method.

**Time and Memory Requirements.** BEiT is trained with a non-contrastive objective and without multi-crop augmentation, thus it consumes only a memory of 5.6G and takes 90.1h for 800 epochs. Comparing iBOT and DINO with multi-crop augmentation, iBOT with MIM induces 25% more memory requirements and 7.4% more actual training time. Considering pre-training efficiency (accuracy versus time), 800-epochs pre-trained DINO requiring for 180.0h, while 300-epochs iBOT only requires 73.3h with 0.4% higher linear probing accuracy (77.0 versus 77.4).

## F    ALTERNATIVE TOKENIZERS

To investigate how different approaches to tokenize the patches affect MIM, we study several alternatives. In BEiT (Bao et al., 2021), masked patches are tokenized by a DALL-E encoder. MPP

(Dosovitskiy et al., 2021) tokenizes the masked patches using their 3-bit mean color. For Patch Clustering, we first perform $K$-Means algorithm to the flattened color vector of each $16 \times 16$ patch ($d = 768$). 10% data of ImageNet-1K training set is sampled and clustered. We set $K$ to 4096. During pre-training, each patch is tokenized by the index of its closest centroids. Lastly, we use 300-epoch pre-trained DINO as a standalone tokenizer. Each patch can be tokenized by the argmax of its output from the pre-trained DINO. We use average pooling to aggregate the patch representations. From Tab. 20, we see that all methods achieve decent fine-tuning results compared to the supervised baseline, while only methods tokenized by semantically meaningful tokenizer have proper results on $k$-NN and linear classification. MPP (Dosovitskiy et al., 2021) and patch clustering rely purely on offline statistics without the extra stage of online training. We find patch clustering has slightly better performance in all three protocols compared to MPP, suggesting the benefits brought by visual semantics. While BEiT has poor $k$-NN and linear probing accuracy, a good fine-tuning result also suggests relatively low requirements for fine-tuning protocol on high-level semantics.

## G  VISUALIZATION

In this section, we first give more visualized pattern layouts and self-attention maps. Beyond that, we consider an additional task of mining sparse correspondences between two images and illustrating the superiority of ViTs by showcasing several visualized results.

### G.1  PATTERN LAYOUT

**Pattern Layout for Patch Tokens.** To illustrate versatile, interesting behaviors iBOT has learned, we organize the visualization of pattern layout in two figures. In Fig. 13, we mainly showcase additional pattern layouts that share high-level semantics. In Fig. 14, we mainly showcase additional pattern layouts that share low-level details like color, texture, shape, *etc*. Top 100 patches with the highest confidence over the validation set are visualized with a $5 \times 5$ context around each $16 \times 16$ patch token (colored orange).

**Composing Images with Representative Patterns.** In Fig. 15, we visualize 4 patches with the highest self-attention score (with non-overlapped assigned index) and also show the pattern layout of that assigned index. The visualized results indicate iBOT can only be represented by several representative patches, which helps the model's robustness and performance in recognition. This is also validated by our part-wise linear probing experiments.

**Comparison with Other Methods.** We visualize pattern layout for patch tokens using other self-supervised methods (Bao et al., 2021; Caron et al., 2021) in Fig. 16. For BEiT, the DALL-E encoder generates a discrete number for each patch token. For DINO, we directly use the projection head for `[CLS]` token and generate a 65536-d probability distribution for each patch token. The index with the highest probability is assigned for the token.

**Pattern Layout for `[CLS]` Token.** We here also provide additional visualization of semantic patterns emerge in `[CLS]` token, which is obtained via self-distillation on cross-view images. We also observe similar behavior in DINO since it's not a unique property brought by MIM. In fact, semantics are now believed to emerge as long as a similarity between two distorted views of one image is enforced (Grill et al., 2020; He et al., 2020; Caron et al., 2020; 2018).

### G.2  SELF-ATTENTION VISUALIZATIONS

Similar to the setting of Sec. 4.3.2, we here provided more self-attention map visualization from multiple heads of the last layer in Fig. 18.

### G.3  SPARSE CORRESPONDENCE.

We consider a sparse correspondence task where the overlapped patches from two augmented views of one image, or patches from two images labeled as one class, are required to be matched. The correlation is sparse since at most $14 \times 14$ matched pairs can be extracted with a ViT-S/16 model. We visualize 12 correspondences with the highest self-attention score extracted from iBOT with

ViT-S/16 pre-trained for $800$ epochs. The score is averaged between multiple heads of the last layer. Several sampled sets of image pairs are shown in Fig. 19. We observe empirically that iBOT perform well for two views drawn from one image, nearly matched the majority of correspondence correctly. In the second column, iBOT can match different parts of two instances from the same class (*e.g.*, tiles and windows of two cars) despite their huge differences in texture or color. We observe the DINO also has comparable visualized effects, illustrating the representation pre-trained with self-distillation also suits well for retrieval in a patch-level scale.

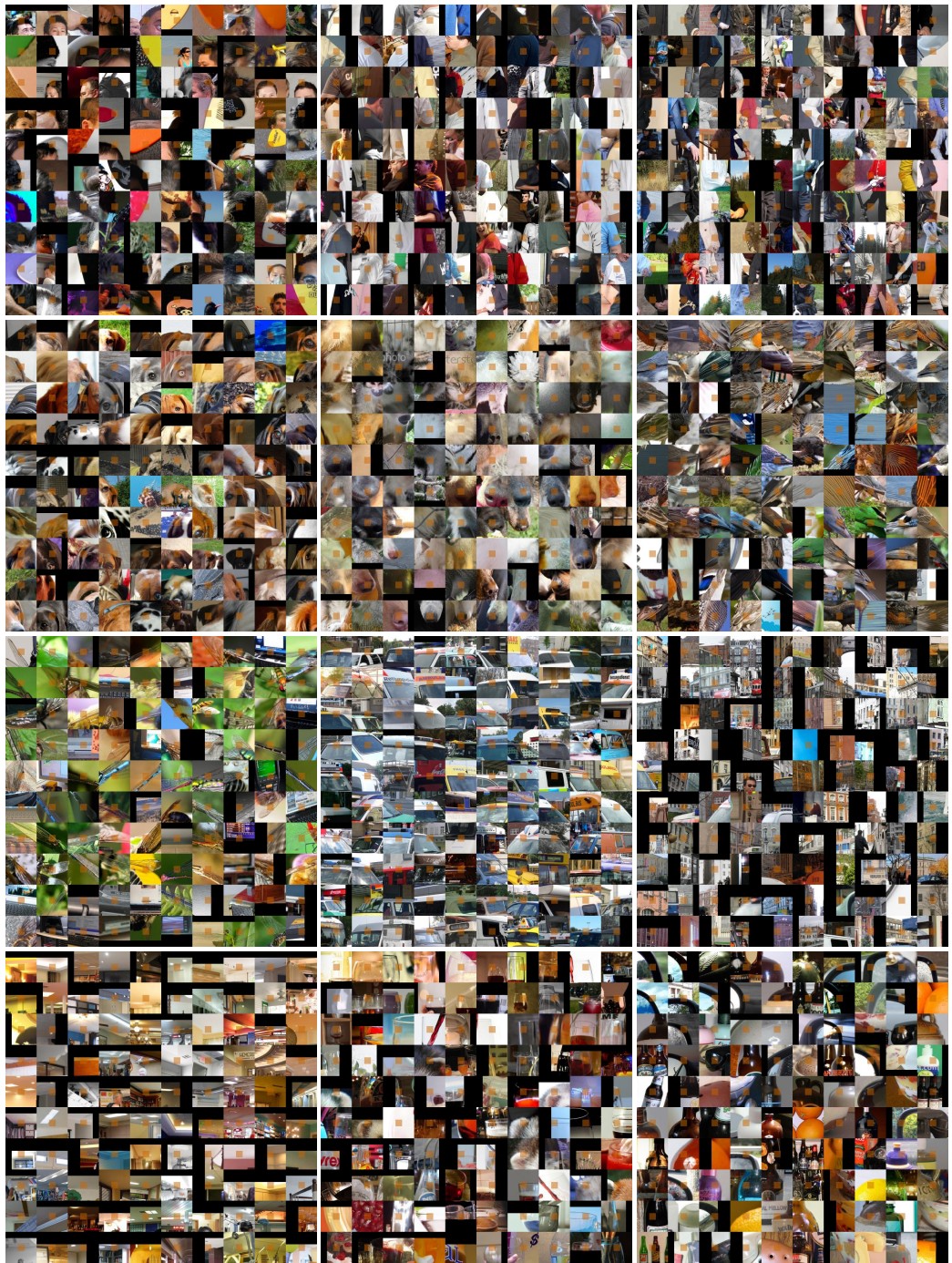

Figure 13: **Visualization for pattern layout of patch tokens that share high-level semantics.** In the first row, we visualize different human-related semantic parts. We observe clear patterns accounting for *human hair*, *human shoulder & arm*, and *human elbow* respectively in the left, middle, and right figure. In the figures from the second row and the left figure from the the third row, we visualize animal-related semantic parts. *dog's ear*, *dog's nose*, *bird's wing*, and *dragonfly's wing* can be observed. In the rest of figures from the third row, we visualize semantic parts related to outdoor scenes. *front window of the vehicle* and *window of the architecture* can be observed. In the last row, we visualize indoor objects like *ceiling* and *glass bottle*.

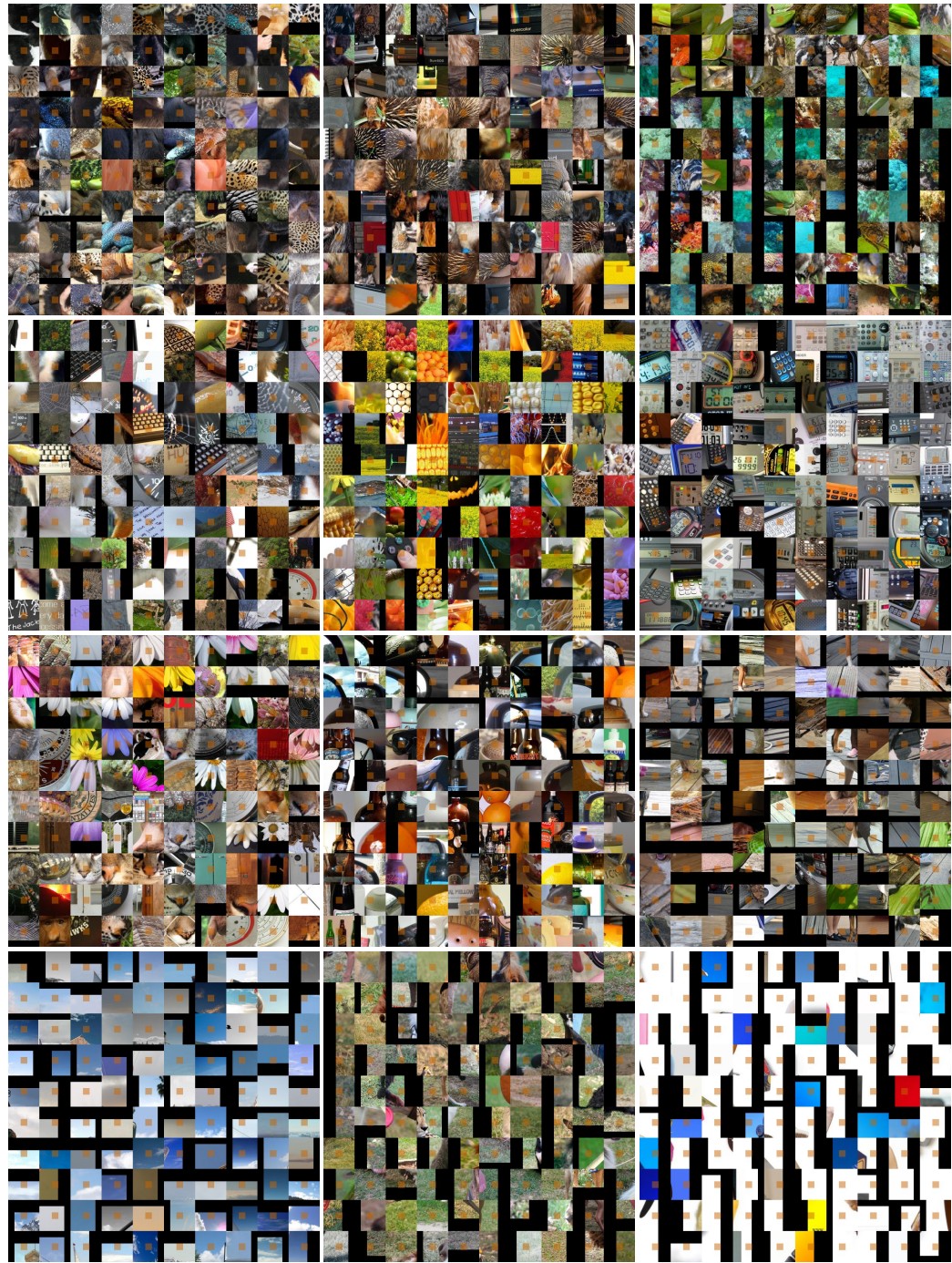

Figure 14: **Visualization for pattern layout of patch tokens that share low-level details.** In the first two columns, we visualize patches that share similar textures. In the first figure, *fur of leopard and the skin of lizard* share a similar dotted texture. In the second figure, *shell of hedgehog and the skin of elephant* share similar striped texture. In the third column, we visualize pattern layouts related to shape. For example, the shape of objects in the left and middle figures share similar curvature. The rightmost patterns clearly depict the shape of a straight line. We visualize pattern layout related to color in the last column, where *blue*, *green* and *white* can be observed.

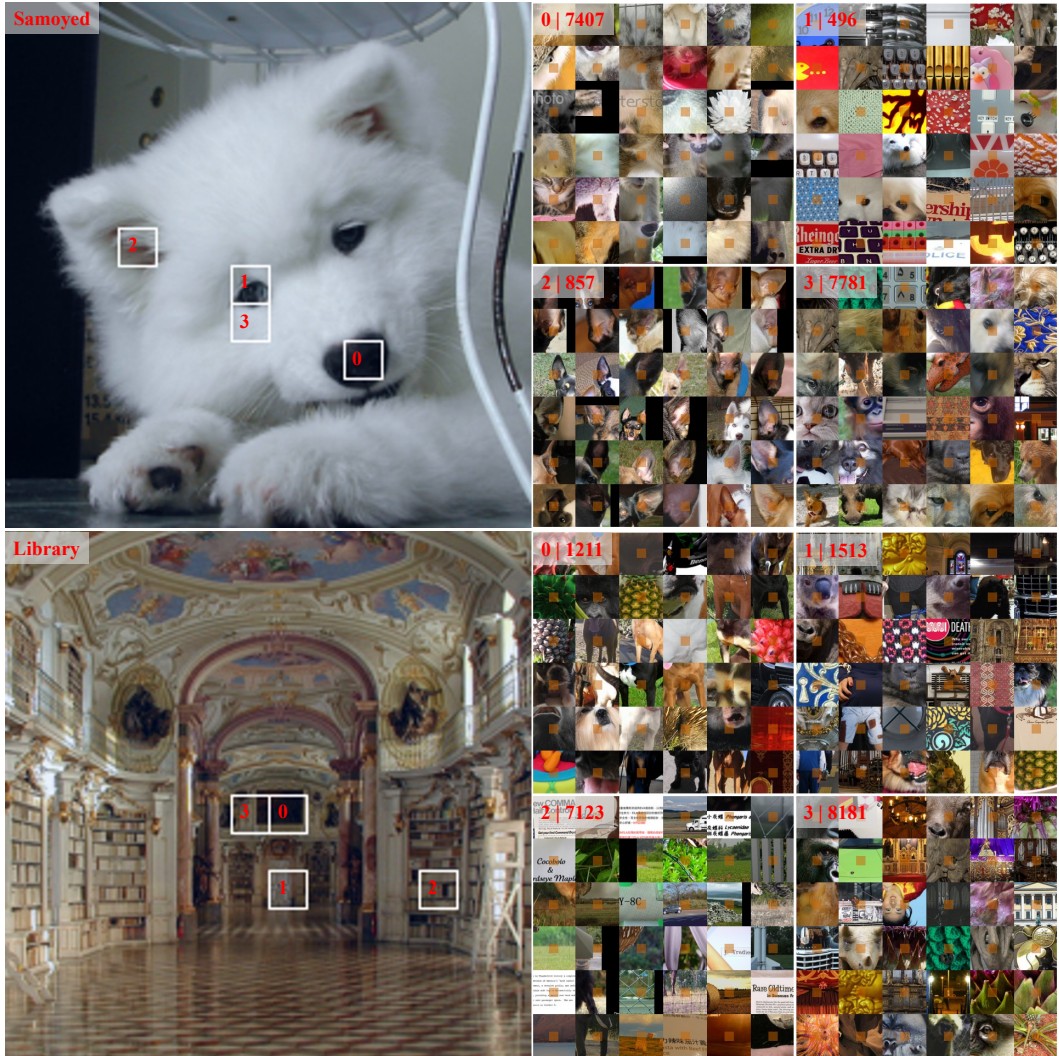

Figure 15: **Top-**4 **representative patches with each of their pattern layout.** Order index 0, 1, 2, 3 are ranked according to its self-attention score. In the top-left corner for each pattern layout subfigure, its order index and cluster index are annotated. In the top panel, we can observe that pattern 0,2,3 show explicit semantic information of *nose, eyes, ears* respectively. Interestingly, patch 1 also locates around the eyes of the *Samoyed* but its corresponding pattern share visual similarity in shape instead of semantics. This illustrate the diverse behaviour for each learned pattern. In the bottom panel, a *library* is represented by 0 *two- or multi-color joints*, 1,3 *knurlling texture*, 2 *texts*. Similarly, we have patterns 0,1,3 focusing more on texture & color and pattern 2 focusing more on semantics. All of these visualized results illustrate versatile behaviour for each index.

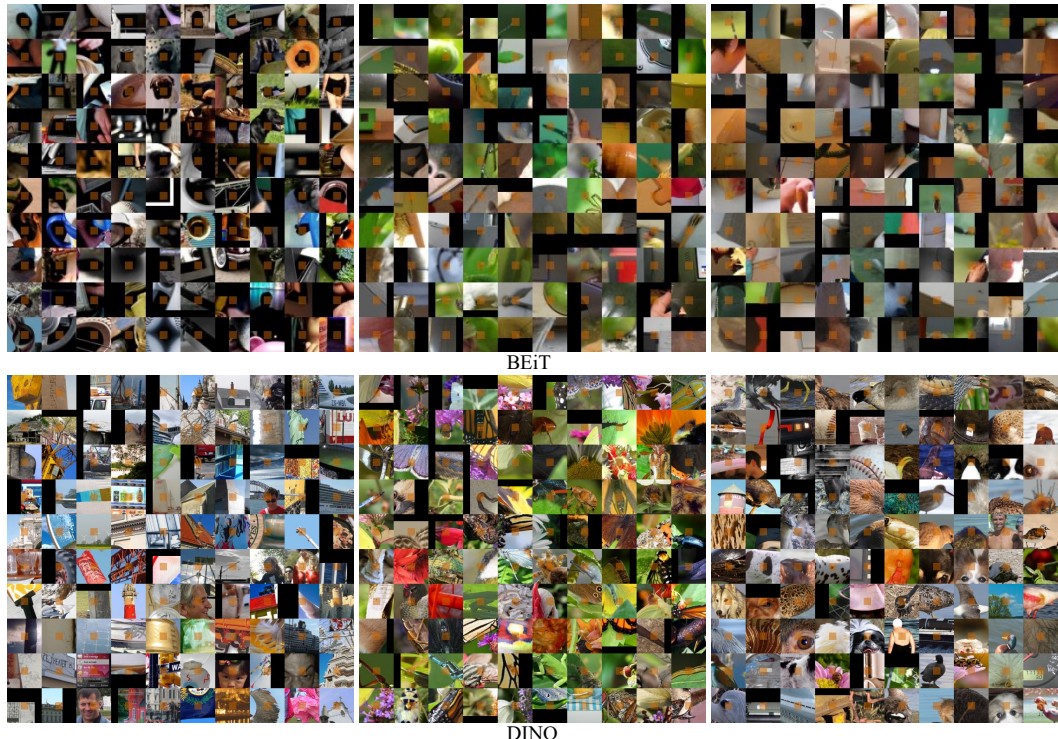

Figure 16: **Visualization for pattern layout of patch tokens using BEiT (top) and DINO (bottom).** In the layout extracted from the DALL-E encoder, we observe minimal semantic patterns. In most cases, patches with similar color (*e.g.*, *black area* in left figure) or texture (*e.g.*, *line* in right figure) are clustered. In the layout extracted from DINO, while more complex textures are visible, most patches share similar local details instead of high-level semantics. In the right figure, the semantic part *eyes* can be somehow observed, yet it is mixed with plenty of irrelevant semantic parts.

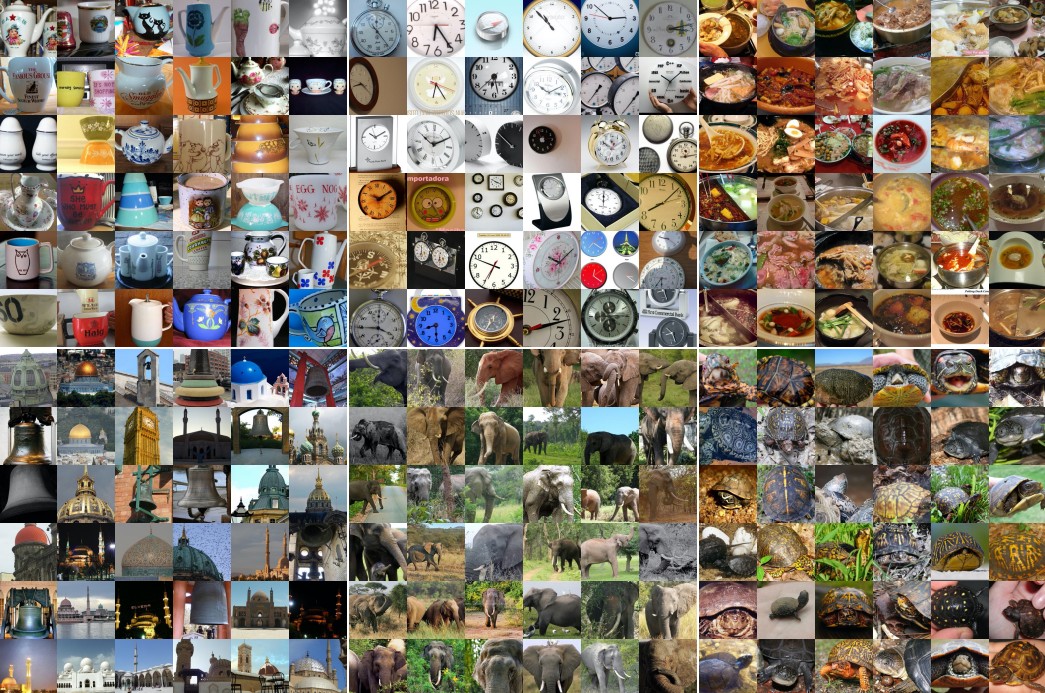

Figure 17: **Visualization for pattern layout of [CLS] token.** We here indicate the high quality of semantic layout brought by self-distillation of cross-view images on [CLS] token. This property is not brought by MIM and is also prominent in DINO.

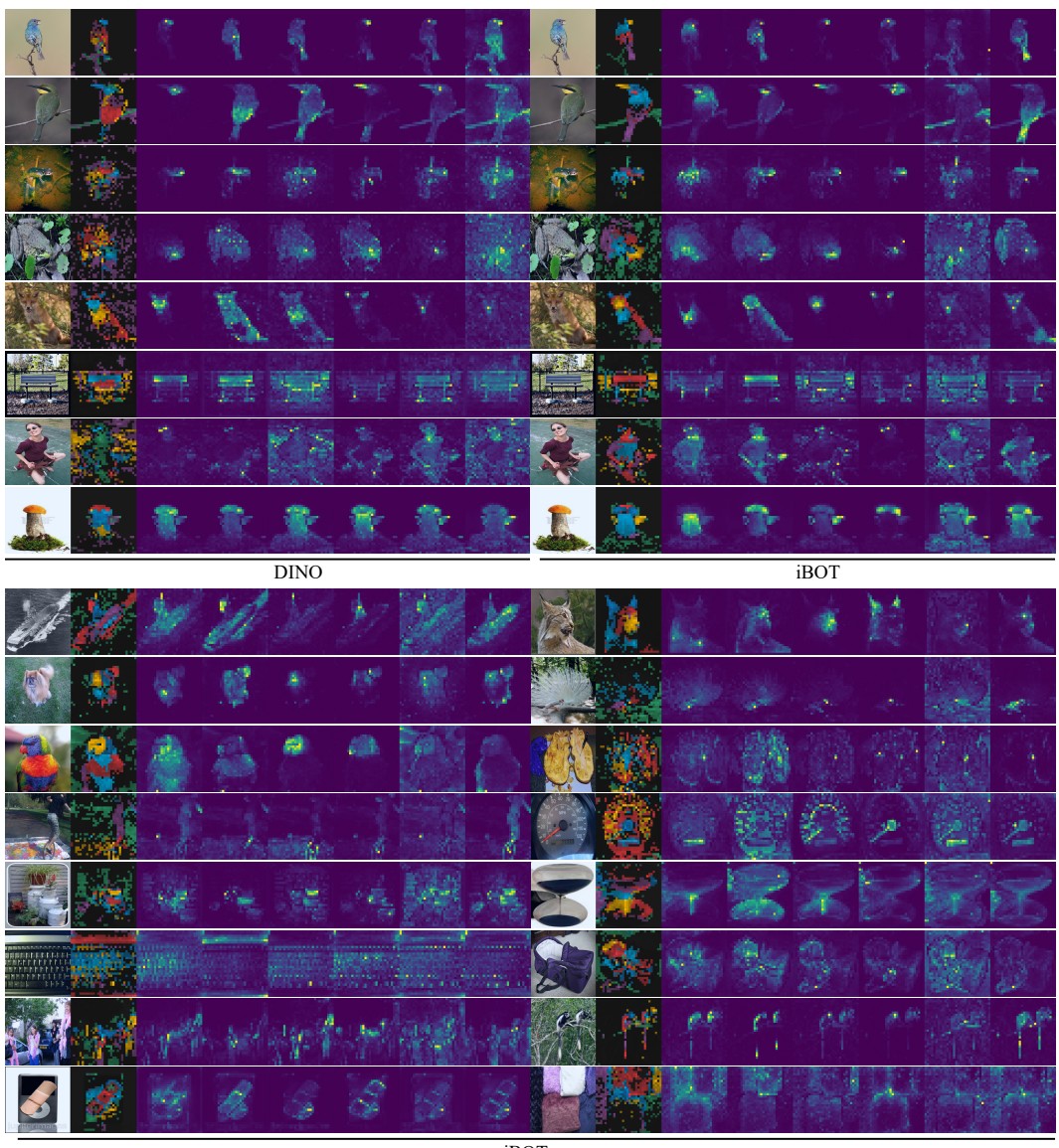

Figure 18: **Visualization for self-attention map from Multiple Heads.** In the first 8 columns, we showcase iBOT's attention map along with DINO's. In the last 10 columns, we showcase more attention map from iBOT. We indicate that iBOT shows visually stronger ability to separate different objects or different parts of one object apart by giving more attentive visualized results for each part, compared with DINO. For example, in the fifth column, there is an attention head in iBOT accounting for the *ear of the fox* solely, while in DINO, it emerges with other parts; In the eighth column, iBOT separates the *mushroom* into more semantically meaningful parts.

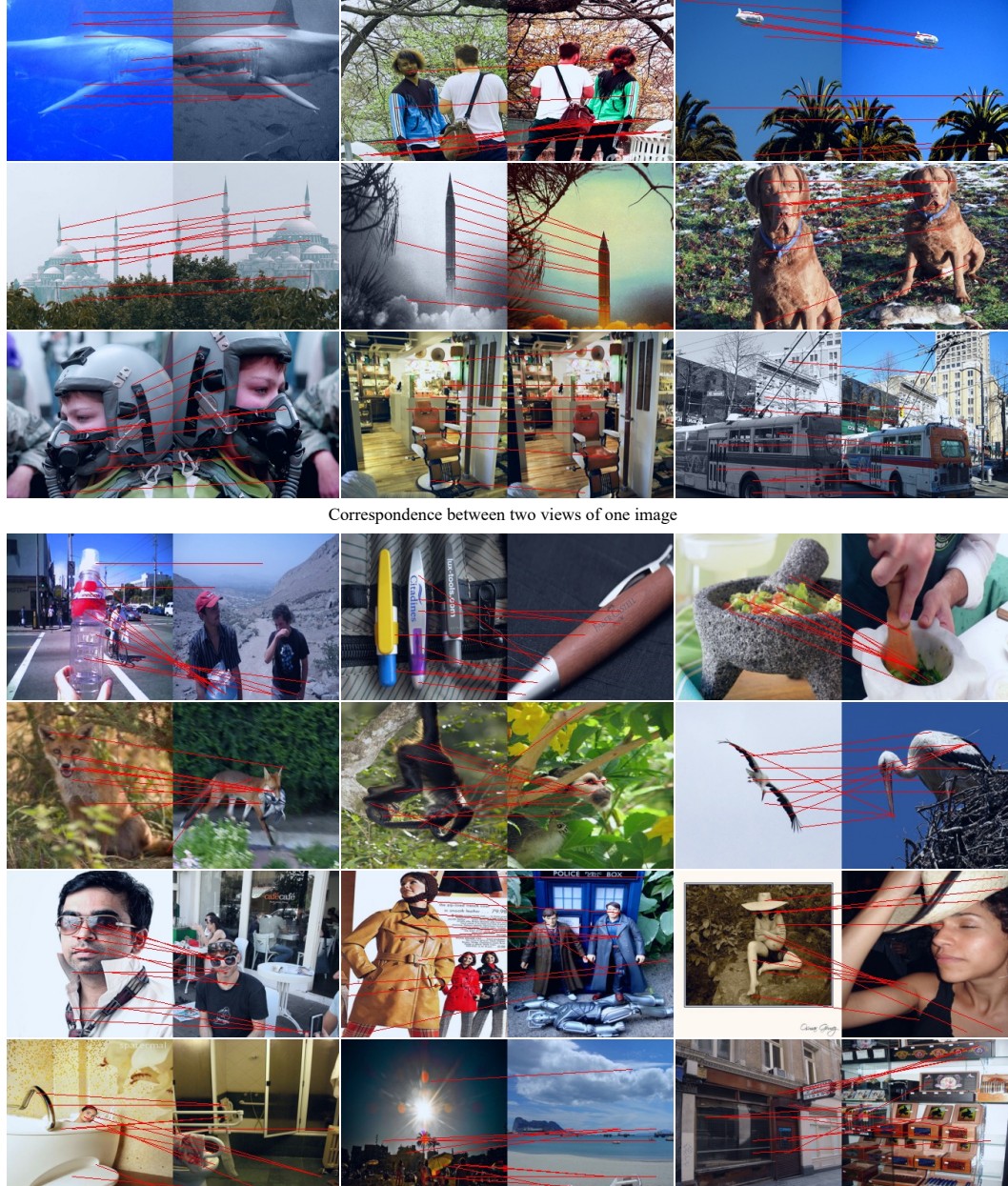

Figure 19: **Visualization for sparse correspondence.** The top panel are images pairs sampled from two views of one image. The extracted correspondence from iBOT is mostly correct despite augmentations on scale and color. The bottom panel are image pairs sampled from two images of one class. The first row is images with salient objects but different sizes, positions and textures. The second row are images draw from animals, and we can observe more clearly that iBOT matches the semantic parts of animals correctly (*e.g.*, *tails of the fox*, *beak of the bird*). The third row is human-centered images with human bodies or clothing. The fourth row is natural or domestic scenes where salient objects are invisible. Although no explicit semantic parts can be matched visible to human's understanding, we can still observe the iBOT can extract correspondence based on their texture or color (*e.g.*, *wooden texture of signboard and boxes*. All these visual results demonstrate strong capability for iBOT in part retrieval or matching in a local scale.

