# OpenReview forum: "Image BERT Pre-training with Online Tokenizer"
_ICLR.cc/2022/Conference — ICLR 2022 Poster_

### Official Review · Reviewer_WgwF · 2021-10-29

**Correctness:** 3
**Technical Novelty And Significance:** 2
**Empirical Novelty And Significance:** 3
**Recommendation:** 6
**Confidence:** 4

**Main Review:**

Strengths:
- **Writting:** The paper is well written
- **Simplicity:** The method is not a significant change with the previous approaches, but the method is simple and give good results.
- **Ablations and Analysis** Provide elements to understand the impact of IBoT modifications compared to a more traditional SSL approach like DINO[1].

Weakness:
- **Comparison with others approaches**: Table 1 and 2, It would be interesting to have a cost comparison of pre-training approaches. Indeed, the number of parameters and the number of epochs used is not very useful. Indeed, using for example multi-crop increases the number of FLOPs and the memory used. Moreover, contrastive approaches using 2 or more different data-augmentation per image in each batch have longer epochs than approaches like BeiT. It would be interesting to try to make a more accurate comparison.

-  **Comparison with supervised baselines for full fine-tuning setting**:  The number of epochs used for the supervised baseline and for the self-supervised approaches are not the same which makes the comparison quite unfair. It would be interesting to compare the performance by adapting the number of epochs in order to better measure the gain compared to the supervised model.

[1] Caron et al., Emerging Properties in Self-Supervised Vision Transformers, ICCV 2021

**Summary Of The Paper:**

The paper proposes the iBOT method. This approach is inspired by the contrastive self-supervised learning approach like DINO and the mask modelling approach like BeiT. The idea is to use an online tokenizer instead of a pretrained tokenizer like Beit. The iBoT approach combines a loss at the patch level like Beit and a global loss at the image level like DINO. IBOT is evaluated in image classification, detection and segmentation.

**Summary Of The Review:**

The paper is quite well written, the method is simple and gives better performance than DINO. Ablations and analysis are quite interesting. Nevertheless, the comparison between the different approaches does not seem to be complete enough. So, it is difficult to see if the iBoT approach brings a significant gain.

---

> ### Author Response · Authors · 2021-11-15
> **Response to Reviewer WgwF**
>
> Thanks for your valuable comments. To address your major concerns around training cost, we first list the time and memory requirements for different methods that are missing in **G1** in our general response. We also define the Effective Training Epochs $E$ in **G1** to account for differences of actual seen images/views during pre-training and conduct a more accurate comparison in the required two aspects.
>
> > **Q1: Longer pre-training epochs for non-contrastive SSL methods.**
>
> Using Effective Training Epochs $E$, we conduct a more fair comparison among methods with different pre-training epochs $P$ and fine-tuning epochs $F$. We pre-train BEiT [1] for 1600 epochs. We indicate that longer-epoch pre-training does not necessarily induce better performance, as shown by little performance gain when increasing the pre-training epochs for BEiT from 300 to 800 epochs. This is probably due to the pretext task of MIM from a pre-trained DALL-E tokenizer is easier to optimize thus requires less data than the contrastive objectives. We also include the fine-tuning results of iBOT w.o.multi-crop, which has the same $E$ with 800-epoch pre-trained BEiT and 400-epoch pre-trained MoCov3 [2], we empirically find that iBOT without multi-crop produces sub-optimal results. Note that MoCov3 satisfies at 300 epochs with an 83.2 top-1 accuracy, also indicating longer training scheduler does not necessarily yield improvement.
>
> | ViT-B/16 | $P$  | $F$ | $E$ | Top-1 Acc |
> | -----    | ---- | ----| ----|----        |
> | BEiT     | 300  | 100 | 400 | 82.8       |
> | BEiT     | 800  | 100 | 900 | 83.2       |
> | BEiT     | 1600 | 100 | 1700| 83.3     |
> | MoCov3     | 300 | 100 | 700 | 83.2      |
> | MoCov3     | 400 | 100 | 900 | 83.2      |
> | iBOT w.o. multi-crop | 400  | 100 | 900 | 82.6 |
> | iBOT     | 400  | 100 | 1700| 83.8       |
>
> > **Q2: Longer fine-tuning epochs for supervised baselines.**
>
> To compare the supervised baseline in fine-tuning, we conducted experiments of randomly-initialized supervised baseline (Rand. sup.) and iBOT self-supervised (iBOT SSL) ViT with longer training epochs. The results in the following table (& Tab. 10) suggest that the supervised model saturates at 300 epochs, which has been observed in Fig 3. of DeiT [3]. The results also reveal that a longer training schedule can even harm the performance for iBOT self-supervised models, due to training saturation and overfitting.
>
> | ViT-B/16   | $P$ | $F$ | $E$ | Top-1 Acc  |
> | -----------| --- | --- | ----| -----------|
> | Rand. sup. |  -  | 300 | 300 | 81.8       |
> | Rand. sup. |  -  | 1700| 1700| 81.8    |
> | iBOT SSL   | 50  | 100 | 300 | 83.0       |
> | iBOT SSL   | 400 | 100 | 1700| 83.8       |
> | iBOT SSL   | 400 | 200 | 1800| 83.2       |
>
>
> ---
>
> We would be more than happy to discuss any further questions!
>
> [1] *Hangbo Bao, Li Dong, and Furu Wei. BEiT: BERT pre-training of image  transformers. Technical report, arXiv:2106.08254.*
>
> [2] *Xinlei Chen, Saining Xie, and Kaiming He. "An empirical study of training self-supervised visual transformers." Technical report, arxiv:2104.02057.*
>
> [3] *Hugo Touvron, Matthieu Cord, Matthijs Douze, Francisco Massa, Alexandre Sablayrolles, and Herve Jegou. Training data-efficient image transformers & distillation through attention. In ICML, 2021.*

---

> > ### Comment · Reviewer_WgwF · 2021-11-22
> > **Thanks for the answer**
> >
> > Thanks for the detailed answer. The rebuttal addresses my main concerns regarding the comparison with other approaches. I will adjust my rating accordingly. Moreover, I think this is a paper with interesting results and insights.

---

### Official Review · Reviewer_s5wd · 2021-10-31

**Correctness:** 3
**Technical Novelty And Significance:** 2
**Empirical Novelty And Significance:** 3
**Recommendation:** 6
**Confidence:** 4

**Main Review:**

### Strength
- The MIM for self-supervised feature learning is well motivated and technically sound.
- Noticeable improvement on downstream tasks like detection and segmentation over the strong baseline DINO with the proposed MIM loss.
- Extensive experimental results and visualization are given to show the effectiveness of the proposed mask image modeling objective and iBoT model. The authors also ablate different tokenization methods and different loss components to validate the contribution.
- Detailed results and training settings are listed which can be helpful for future research.

### Weaknesses
- The improvement of the fine-tuned model in Tab 2 is limited. It would be better if the authors could explain more on this part.

- The experimental results on the downstream tasks look good and the authors also provide some insights, but the technical contribution is limited.

- The main difference between DINO and proposed iBoT is 1) masked image as input for the student model; 2) MIM loss for self-distillation on reconstructed patch tokens. Table 8 and section E in the appendix only provide ablation on the tokenization method, loss function, and dense self-distillation. To be fully convinced by the MIM loss, I think the author should also provide the result of the iBoT model without MIM loss (i.e. use the masked image as input for the student model of DINO). Since applying the mask on the input image is similar to Cutout augmentation, it can also improve the robustness and the performance of the trained model.

- It would be better if the authors provide more comparisons on the proposed online tokenizer and pre-trained tokenizer (as in BEiT). It's not clear to me why the proposed online tokenizer can better capture the semantically meaningful pattern.

- As multi-crop augmentation is important for both Swav and DINO, I think it’s also important to explain and analyze more on how to apply multi-crop augmentation with mask image modeling in the main paper. Also, the training costs (actual training time) are not listed. Is there any additional training cost for applying masked image modeling?


**Summary Of The Paper:**

This paper presents a new strategy for self-supervised training of vision transformer models by combining the self-distillation-based method DINO with mask image modeling. The momentum teacher model in DINO is taken as an online tokenizer. A mask image modeling loss that computes self-distillation loss between the reconstructed tokens and the output tokens of the momentum teacher is proposed. Extensive experiments are conducted to show the effectiveness of the proposed self-supervised training framework. The result on downstream tasks including classification, object detection, and segmentation also shows the benefit of the mask image modeling loss.

**Summary Of The Review:**

Overall, I think this is a practical method with good experimental results. However, the proposed mask image modeling loss may not be clearly verified. As its current state, I would like to rate this paper as marginally below the acceptance threshold.

---

> ### Author Response · Authors · 2021-11-15
> **Response to Reviewer s5wd**
>
> Thanks for your valuable comments. Below we discuss the points you have raised in detail.
>
> > **Q1: Limited improvement under fine-tuning protocol in Tab. 2.**
>
> We think limited performance is mainly due to performance being close to saturation under the settings of pre-training and fine-tuning on the same dataset, ImageNet-1K. In practice, the effect of pre-training is revealed especially when the size of pre-training data greatly exceeds the fine-tuning data. We show the number here that the performance gain increases under a semi-supervised setting as the finetuning data becomes less. To further validate, we provide additional fine-tuning results with ImageNet-22K as pre-training data. With ViT-B/16, the performance gain of iBOT over BEiT increases from 0.4 to 0.7, suggesting iBOT benefits from pre-training with larger-scale data.
>
> ViT-S/16:
>
> |Pre-Train|Fine-Tune|DINO|iBOT|Gain|
> |-|-|-|-|-|
> |ImageNet-1K|ImageNet-1K|82.0|82.3|+0.3|
> |ImageNet-1K|10% of ImageNet-1K|74.3|75.1|+0.8|
> |ImageNet-1K|1% of ImageNet-1K|60.3|61.9|+1.5|
>
> ViT-B/16:
>
> |Pre-Train|Fine-Tune|BEiT|iBOT|Gain|
> |-|-|-|-|-|
> |ImageNet-1K|ImageNet-1K|83.4|83.8|+0.4|
> |ImageNet-22K|ImageNet-1K|83.7|84.4|+0.7|
>
> > **Q2: Limited technical contributions.**
>
> Thanks for appreciating the effectiveness of our work. We discuss our technical novelty and contributions in **G2** in our general response.
>
> > **Q3: DINO with the masked image as augmentation.**
>
> We conducted experiments using more sophisticated augmentation in our early trials (50-epoch pre-training, w.o. multi-crop) and find that this will instead intervene in the pre-training and harm the downstream performance. While augmentation like masking can improve the performance of supervised models, we do not observe the behavior of "improve the robustness and the performance of the trained model"  when applying image masking or random erasing to self-supervised pre-training.
>
> |Ablations|$K$-NN|
> |-|-|
> |DINO|52.1|
> |DINO w. masked student input, ratio=0.1|48.8|
> |DINO w. masked student input, ratio=0.3|41.1|
> |DINO w. Random Erasing, prob=0.25|43.5|
>
> With the above empirical results, we here briefly analyze why masked input along does not induce better classification accuracy:
> - The augmentation of random erasing or image masking can be too strong for the contrastive objective. Therefore, inter-class features can be potentially collapsed to the same point since the model is able to cheat with a shared mask representation applied to all the images. An interesting reference is [1], and in this case, the Overly Overlapping among different semantic contents may occur, where the inter-class features become mixed and inseparable.
> - Without MIM loss, the model may not explicitly model the internal image structure, thus loses the part-level semantics that is deemed as the major reason for better image recognition performance of iBOT.
>
> > **Q4: Pre-trained DALL-E tokenizer versus online tokenizer.**
>
> We here clarify the difference between the pre-trained DALL-E tokenizer (used in BEiT [2]) and the online tokenizer (used in iBOT) by analyzing both qualitatively and quantitatively.
> - Quantitatively, we indicate that the DALL-E tokenizer itself obtains only a K-NN accuracy of 2.4% and linear probe accuracy of 10.6%, suggesting minimal semantic patterns are captured.
> - Qualitatively, we visualize that the pattern layouts extracted BEiT is mostly low-level details, like color block and textures (the upper part of Fig. 16).
>
> |ViT-S/16|$K$-NN|Linear|
> |-|-|-|
> |DALL-E tokenizer|2.4|10.6|
> |BEiT|6.9|23.5|
> |iBOT / Online tokenizer|69.1|74.2|
>
> The reason why online tokenizer captures better semantic patterns lies intrinsically in its multi-tasking with a contrastive objective, thus the semantic patterns captured by online representation bootstrapping can be transferred to MIM via a shared backbone and projection head. In comparison, the pre-trained DALL-E tokenizer is supervised by a reconstruction objective, which does not guarantee to learn class-discriminative features as desired in the downstream classification.
>
> > **Q5: MIM with multi-crop should be mentioned in the main text.**
>
> Thanks for the suggestions. We will add more discussions of random MIM for iBOT with multi-crop augmentation in the main text. We would also to mention that while multi-crop augmentation plays a crucial role for iBOT pre-training, it is not necessary. iBOT w.o. multi-crop can achieve a linear probing accuracy of 76.2% surpassing 73.4% achieved by MoCov3, as reported in Tab. 9 of our paper.
>
> > **Q6: Training costs.**
>
> For comparison of training costs among different methods, we recommend the reviewer to refer to **G1** in our general response.
>
> ---
>
> We would be more than happy to discuss any further questions!
>
> [1] *Anonymous. Chaos is a Ladder: A New Understanding of Contrastive Learning. Submitted to ICLR, 2022.*
>
> [2] *Hangbo Bao, Li Dong, and Furu Wei. BEiT: BERT pre-training of image  transformers. Technical report, arXiv:2106.08254.*

---

> > ### Comment · Reviewer_s5wd · 2021-11-20
> > **Thanks for the clarification**
> >
> > Thank you for the new results and the detailed feedback. The response to Q3 addressed my main concern about the proposed method. The results under the semi-supervised setting also help to demonstrate the effectiveness of the iBOT model. Overall, I believe this is a good paper with some insight and good results. Therefore, I'd like to upgrade my rating.

---

### Official Review · Reviewer_RoXr · 2021-11-03

**Correctness:** 4
**Technical Novelty And Significance:** 3
**Empirical Novelty And Significance:** 3
**Recommendation:** 8
**Confidence:** 4

**Main Review:**

The paper is clearly written, I believe I could reimplement the method from the paper (using the appendix).
It contains plenty of ablations that answer most of my questions about variants and details of the model.
The experimental results are (very) competitive, the comparisons are relevant, the litterature is adequately discussed.

Maybe a missing interesting comparison would have been BYOL with a VIT model.
The visualizations are convincing (in the main body but also in the appendix), although the weaker point (which is not a major flaw) of the paper is probably section 4.3.1, where it is not very clear how the learned patterns differ really from BEiT or DINO, and more to the point, how the help from this training procedure in image recognition is of any different kind than from the competition.


**Summary Of The Paper:**

This is one of the first papers doing BERT-style pretraining for vision transformers, in particular the first to not use a pre-trained tokenizer (here the (part of) the network that generates the targets for the masked tokens). This has a similarity with BEiT, although BEiT uses an externally procured tokenizer (CLIP, so trained with labels). Instead, here, the idea is to use a teacher (EMA from student) to propose the targets for the masked tokens, yielding a masked image modeling task (L_MIM). The whole model and approach works with 2 augmented views of the same image (as in BYOL or DINO) and also has a self-distillation loss (L_CLS). L = L_MIM + L_CLS. The paper presents state of the art results in kNN (retrieval), image classification with linear probing and fine-tuning (ImageNet), image detection, instance segmentation (both on COCO), and semantic segmentation (ADE20K), within the classes of models that the authors studied.


**Summary Of The Review:**

This is a well presented paper that shows convincing progress in masked image modeling pretraining for vision transformers. It should be accepted for publication at ICLR.

---

> ### Author Response · Authors · 2021-11-15
> **Response to Reviewer RoXr**
>
> Thanks for your valuable comments and your appreciation for our technical contributions. Below we discuss the points you have raised in detail.
>
> > **Q1: BYOL [1] with a ViT model.**
>
> Thanks for your suggestions. In DINO [2] paper, the authors compared DINO with BYOL with ViT-S/16. We here include iBOT in the comparison for completeness.
>
> | Method | Multi-Crop | $K$-NN | Linear |
> | -------| -------| -------| -------|
> | BYOL   | $\chi$ | 66.6   | 71.4   |
> | DINO   | $\chi$ | 67.9   | 72.5   |
> | iBOT   | $\chi$ | 70.3   | 74.8   |
> | BYOL   | $\checkmark$ | 59.8   | 64.8   |
> | DINO   | $\checkmark$ | 72.7   | 75.9   |
> | iBOT   | $\checkmark$ | 74.6   | 77.4   |
>
> > **Q2: Difference of visualized pattern layout with BEiT [3] and DINO.**
>
> Due to space limitations, we have provided the visualized pattern layouts of BEiT and DINO in Fig. 16 of the paper. We identify that:
>
> - BEiT is clearly unable to capture high-level semantic parts, focusing mostly on local-level details like color blocks and line textures. As for the reason, BEiT is tokenized by a DALL-E encoder, which is trained with a reconstruction loss and learns insufficient class-discriminative features. This can be directly shown by the poor $K$-NN and linear probing accuracy of BEiT model.
> - While DINO showcases more complicated patterns, such ability is clearly less performant than iBOT with many irrelevant contents within each cluster. Therefore, DINO w.o. MIM objective may not explicitly model images' internal structure.
>
> In contrast, iBOT produces reliable semantic-meaningful pattern layouts, where the human can assign clear semantic meaning to all patches belonging to the same pattern, as shown in Fig. 4. The visualization shows iBOT helps the model to learn class-discriminative part-level features, which is not observed from the competitors of our training procedure.
>
> ---
>
> We would be more than happy to discuss any further questions!
>
> [1] *Jean-Bastien Grill, Florian Strub, Florent Altch ́e, Corentin Tallec,  Pierre Richemond, Elena Buchatskaya, Carl Doersch, Bernardo Avila Pires, Zhaohan Guo, Mohammad Gheshlaghi Azar, Bilal Piot, koray kavukcuoglu, Remi Munos, and Michal Valko. Bootstrap your own latent: A new approach to self-supervised learning. In NeurIPS, 2020.*
>
> [2] *Mathilde Caron, Hugo Touvron, Ishan Misra, Herv ́e J ́egou, Julien Mairal, Piotr Bojanowski, andArmand Joulin. Emerging properties in self-supervised vision transformers. In ICCV, 2021.*
>
> [3] *Hangbo Bao, Li Dong, and Furu Wei. BEiT: BERT pre-training of image  transformers. Technical report, arXiv:2106.08254.*

---

### Author Response · Authors · 2021-11-15
**General Response [1/2]**

We thank all the reviewers for their valuable comments and insightful advice. Below we address several common concerns and list the supplemented experiments.

> **G1: Training Costs.**

We detail the actual training time (T) and GPU memory (Mem.) of different methods. All methods are trained on two 8-GPU V100 machines with a batch size of 1024.

|ViT-S/16|Crop Number|T$_{100}$|T$_{300}$|T$_{800}$|Mem.|Lin.$_{300}$|Lin.$_{800}$|Fin.$_{800}$|
|-|-|-|-|-|-|-|-|-|
|BEiT|$1\times224^2$|11.3h|33.7h|90.1h|5.6G|20.7|24.2|81.4|
|DINO|$2\times224^2$|15.1h|44.7h|111.6h|9.3G|72.5|73.7|80.8|
|iBOT|$2\times224^2$|15.6h|47.0h|126.4h|13.1G|74.8|76.2|81.6|
|DINO|$2\times224^2+10\times 96^2$|24.2h|72.6h|180.0h|15.4G|76.2|77.0|82.0|
|iBOT|$2\times224^2+10\times 96^2$|24.3h|73.3h|193.4h|19.5G|77.4|77.9| 82.3 |

Comparing iBOT and DINO [1] with multi-crop augmentation, iBOT with MIM induces 25% more memory requirements and 7.4% more actual training time compared to DINO since we compute each patch token through the projection head. However, we highlight that:
- While MIM methods (like BEiT [2]) or contrastive methods without multi-crop (like MoCov3 [3]) require less training time, using a longer training schedule (with more epochs) does not necessarily bring better downstream results. The reported results in the paper are ensured to be close to saturation.
- Considering pre-training efficiency (accuracy versus time), 800-epochs pre-trained DINO requiring for 180.0h, while 300-epochs iBOT only requires 73.3h with 0.4% higher linear probing accuracy (77.0 versus 77.4). This clearly suggests iBOT can obtain higher accuracy with less training time. Although ~25% more memory consumption is imposed, we think it's bearable since iBOT is trained with a reasonable batch size of 1024.

Due to extra computation costs brought by multi-crop augmentation, different methods with the same pre-training epochs $P$ and fine-tuning epochs $F$ actually see different total numbers of images. To mitigate, we propose to measure the Effective Training Epochs $E$, defined as $E = P\times r+F$, where $r$ is a scaling factor accounting for actual trained images/views of different resolutions. DINO and iBOT are by default trained with $2$ global crops of size $224 \times 224$ and $10$ local crops of size $96\times96$. Thus $r=2+(\frac{96}{224})^2\times 10=3.84\approx4$ for DINO and iBOT, while $r=1$ for BEiT. For those who are interested, we detail the discussions and comparisons in our response to **R3**.

> **G2: Technical contributions.**

- We indicate the promising potential of BERT-like pre-training for Vision Transformers and indicate the advantage of using a semantic-meaningful visual tokenizer.
- We present a novel self-supervised framework, iBOT, via performing self-distillation for MIM with the help of twin teacher as online tokenizer, whose semantic emerges progressively by bootstrapping online global representation.
- We identify an emerging part-level semantics induced by MIM, that can help the model with image recognition both on global and local scales. iBOT can achieve a linear probing accuracy of 79.5% and an mAP of 50.0% on semantic segmentation with ViT-B/16.
- We conduct extensive analysis (Sec. 4.3) and ablations (Sec. 4.4) to help understand the impact of MIM compared with traditional SSL approaches (e.g., BEiT and DINO).
- We find that MIM pre-trained Vision Transformers scale well to large-scale data and model size, which largely closes the gap between masked modeling between language and vision. With ViT-L/16 and ImageNet-22K pretraining, iBOT achieve a linear probing accuracy of 81.7% and a fine-tuning accuracy of 86.3%.
- We believe that our method is "well-motivated and technically sound" (**R2**), "simple and give good results" (**R3**), and "shows convincing progress in masked image modeling" (**R1**). Most importantly, iBOT "is one of the first papers doing BERT-style pre-training for Vision Transformers" (**R1**).

We hope that iBOT brings new perspectives to the study of self-supervised visual pre-training and opens new research directions.

---

> ### Author Response · Authors · 2021-11-15
> **General Response [2/2]**
>
> > **G3: Using Swin-Transformer as the backbone.**
>
> To validate the effectiveness of iBOT, we provide results of iBOT trained with Swin-Transformer. Under the linear probing protocol, Swin-T with a window size of 7 and 14 achieves a top-1 accuracy of 78.6% and 79.3% respectively, surpassing the previous best result, 78.1% and 78.7% by EsViT [4].
>
> |Method|Arch.|$K$-NN|Linear|
> |-|-|-|-|
> |EsViT|Swin-T/7|75.7|78.1|
> |EsViT|Swin-T/14|77.0|78.7|
> |iBOT|Swin-T/7|75.3|78.6|
> |iBOT|Swin-T/14|76.2|79.3|
>
> > **G4: Pre-training with ImageNet-22K.**
>
> To illustrate the idea of MIM can help scale pre-training to large-scale data, we provide ImageNet-1k fine-tuning results with models pre-trained with ImageNet-22K. iBOT achieves a top-1 accuracy of 84.4% with ViT-B/16, with a 0.6% performance gain. Compared 0.3% performance gain of more data in BEiT, iBOT can be more scalable to larger-scale data. With ViT-L/16, iBOT achieves an accuracy of 86.3%, surpassing the previous best result, 86.0% by BEiT. Additionally, ImageNet-22k pre-trained ViT-L/16 achieve a linear probing accuracy of 81.7%, surpassing the previous best result, 81.3% with Swin-B/14 by EsViT.
>
> |Arch.|Linear|Finetune|
> |-|-|-|
> |ViT-B/16|79.1|84.4|
> |ViT-L/16|81.7|86.3|
>
> ---
>
> We will update our draft with the above materials. More discussions are welcombed!
>
> [1] *Mathilde Caron, Hugo Touvron, Ishan Misra, Herv ́e J ́egou, Julien Mairal, Piotr Bojanowski, andArmand Joulin. Emerging properties in self-supervised vision transformers. In ICCV, 2021.*
>
> [2] *Hangbo Bao, Li Dong, and Furu Wei. BEiT: BERT pre-training of image  transformers. Technical report, arXiv:2106.08254.*
>
> [3] *Xinlei Chen, Saining Xie, and Kaiming He. "An empirical study of training self-supervised visual transformers." Technical report, arxiv:2104.02057.*
>
> [4] *Chunyuan Li,  Jianwei Yang,  Pengchuan Zhang,  Mei Gao,  Bin Xiao,  Xiyang Dai,  Lu Yuan,  andJianfeng Gao.   Efficient self-supervised vision transformers for representation learning. Technical report, arxiv:2106.09785.*

---

### Author Response · Authors · 2021-11-22
**Summary of Revision**

We thank all reviewers for the valuable comments. We hope our response addresses your concerns and we upload a draft with a few modifications, detailed as follows:

- Add a comparison of training cost (actual training time, GPU memory) among different pre-training methods. (Appendix E) [**R2**,**R3**]
- Provide a more accurate comparison among different methods via Effective Training Epochs. Clarify that a longer training schedule does not necessarily yield improvement. (Tab. 1/2/3) [**R3**]
- Include prior practices of MIM in the field of vision language pre-training to related work. (Sec. 5)
- Include iBOT with Swin-Transformer as the backbone. (Tab. 1)
- Scale the pre-training data of iBOT to ImageNet-22K. (Tab. 3)

We would be more than happy to discuss any further questions!

---

### Decision · Program_Chairs · 2022-01-20

**Decision:**

Accept (Poster)

**Comment:**

The paper is interesting, and its focus is timely and important, given the continuing rapid rise of transformers (and their dependence of tokenization of images). All three reviewers recommend acceptance, to varying degree. The paper will be a valuable contribution to the program at ICLR.